

# A Greenland-wide empirical reconstruction of paleo ice-sheet retreat informed by ice extent markers: PaleoGrIS version 1.0

Tancrède P.M. Leger[1], Christopher D. Clark[1], Carla Huynh[2], Sharman Jones[3], Jeremy C. Ely[1], Sarah L. Bradley[1], Christiaan Diemont[1], Anna L.C. Hughes[4]

[1]Department of Geography, University of Sheffield, Sheffield, S10 2TN, United Kingdom
[2]School of GeoSciences, University of Edinburgh, Drummond Street, Edinburgh, EH8 9XP, United Kingdom
[3]Aberystwyth University, Geography and Earth Sciences, Aberystwyth, SY23 3DB, United Kingdom
[4]Department of Geography, School of Environment, Education and Development, The University of Manchester, Manchester, M13 9PL, United Kingdom

*Correspondence*: Tancrède P. M. Leger (t.p.leger@sheffield.ac.uk): personal address: tankleger@gmail.com

**Abstract.** The Greenland Ice Sheet is a large contributor to global sea-level rise, and current mass losses are projected to accelerate. However, model projections of future ice-sheet evolution are limited by the fact that the ice sheet is not in equilibrium with present-day climate, but is still adjusting to past changes that occurred over thousands of years. Whilst the influence of such committed adjustments on future ice-sheet evolution remains unquantified, it could be addressed by calibrating numerical ice sheet models over larger timescales and, importantly, against empirical data on ice margin positions. To enable such paleo data-model interactions, we need Greenland-wide empirical reconstructions of past ice-sheet extent that combine geomorphological and geochronological evidence. Despite an increasing number of field studies producing new chronologies, such a reconstruction is currently lacking in Greenland. Furthermore, a time-slice reconstruction can help: i) answer open questions regarding the rate and pattern of ice margin evolution in Greenland since the glacial maximum, ii) develop a standardised record of empirical data, and iii) identify understudied sites for new field campaigns. Based on these motivations, we here present PaleoGrIS 1.0, the first Greenland-wide isochrone reconstruction of ice-sheet extent evolution through the Late-Glacial and early-to-mid Holocene informed by both geomorphological and geochronological markers. Our isochrones have a temporal resolution of 500 years and span ~7.5 kyr from approximately 14 to 6.5 kyr BP. We here describe the resulting reconstruction of the shrinking ice sheet and conduct a series of ice-sheet wide and regional analyses to quantify retreat rates, areal extent change, and their variability across space and time. During the Late-Glacial and early-to-mid Holocene, we find the Greenland Ice Sheet has lost about one third of its areal extent (0.89 million km$^2$). Between ~14 and ~8.5 kyr BP, it experienced a near constant rate of areal extent loss of $170 \pm 27$ km$^2$ yr$^{-1}$. We find the ice-sheet-scale pattern of margin retreat is well correlated to atmospheric and oceanic temperature variations, which





implies a high sensitivity of the ice sheet to deglacial warming. However, during the Holocene, we observe inertia in the ice-sheet system that likely caused a centennial to millennial-scale time lag in ice-extent response. At the regional scale, we

35    observe highly heterogeneous deglacial responses in ice-extent evident in both magnitude and rate of retreat. We hypothesise that non-climatic factors, such as the asymmetrical nature of continental shelves and onshore bed topographies, play important roles in determining the regional-to-valley scale dynamics. PaleoGrIS 1.0 is an open-access database designed to be used by both the empirical and numerical modelling communities. It should prove a useful basis for improved future versions of the reconstruction when new geomorphological and geochronological data become available.



# 1 Introduction

The Greenland Ice Sheet holds an estimated ~7.4 m of global sea-level equivalent (Morlighem et al., 2017). It is currently experiencing mass loss at a rate of ~150 Gt yr$^{-1}$ (over the 1992-2018 period; The IMBIE Team, 2019; Otosaka et al., 2023),
making it a large contributor to current global mean sea-level rise. In a warming world, these losses are projected to accelerate (Meredith et al., 2019), with the latest Greenland Ice Sheet Model Intercomparison Project (ISMIP6) estimating a contribution to global mean sea-level rise of 32 ± 17 and 90 ± 50 mm by year 2100 for the RCP2.6 and RCP8.5 greenhouse gas concentrations scenarios, respectively (Goelzer et al., 2020). The robustness of such projections are limited by the short time span of instrumental evidence (10s to 100 years) used for verification and testing (Larsen et al., 2015). This is
problematic because ongoing and future ice sheet mass losses are not exclusively a consequence of contemporary and future climate, but also represent adjustments to climate variations stretching further back in time (100s to 1000s of years; Rogozhina et al., 2011; Yang et al., 2022). As few models include the committed response to late-Quaternary environmental changes, current projection simulations might effectively be starting from the wrong state, whose influence is thus far unquantified. Consequently, improved empirical reconstructions of late-Quaternary ice sheet evolution may be useful for
refining future predictions of mass loss and global sea-level rise, by enabling the calibration of ice sheet models over larger timescales encompassing thousands of years (e.g. Lecavalier et al., 2014; Albrecht et al., 2020).

The response of the ice sheet to climate change during the Late-Quaternary likely contributes to major imbalances in its current state (Calov & Hutter, 1996), yet many key questions pertaining to this time remain unresolved. The position, rate,
and pattern of ice margin retreat during the Late-Glacial and early-to-mid Holocene periods (between ~14 and ~6 kyr BP) remains poorly constrained. During that time, the ice sheet was responding to rapid and high-amplitude fluctuations in atmospheric and oceanic boundary conditions (e.g. the Bølling-Allerød Interstadial; Buizert et al., 2018), whilst also adjusting to fast rates of relative sea level change following the demise of other ice sheets (e.g. during Meltwater Pulse 1A; Lin et al., 2021). Consequent changes in ice sheet margin, volume, and surface mass balance were likely associated with
modifications to the configurations of ice divides and to the position and discharge regimes of major ice streams (e.g. Franke et al., 2022). A key question concerns how far the ice sheet retreated behind its contemporary margin in response to the warming of the Holocene Thermal Maximum (10-5 kyr BP; Cartapanis et al., 2022), a possible analogue to atmospheric warming expected in future decades (Funder et al., 2011). Calibrating model simulations against known Holocene margin positions around the full ice sheet perimeter might help answer this latter question.

The Greenland Ice Sheet has been subject to an increasing number of field studies reconstructing the deglacial evolution of the former ice-sheet margin, both on land and offshore. These typically use geomorphological, sedimentological, and geochronological dating analyses (e.g. Bennike & Weidick, 2001; Hughes et al., 2012; Briner et al., 2014; Larsen et al.,



2014; Young et al., 2021; Garcia-Oteyza et al., 2022; Sbarra et al., 2022). This growing library of geological evidence
enables an improved understanding of the ice sheet response to deglacial climate change, at both regional and continental
scale (Sinclair et al., 2016). However, to our knowledge, it appears that a Greenland-Ice-Sheet wide reconstruction with an
open-access, reproducible database of geomorphological and geochronological evidence is lacking. Such a product would
prove useful to both the modelling and empirical glaciology communities. It would enable ice-sheet scale analyses, help
keep a standardised record of dating investigations, facilitate targeting of understudied regions for future fieldwork, and
provide a reconstruction for calibrating ice sheet model simulations. We note that ice-sheet wide summary datasets of
accumulated evidence have played an essential and enabling role in advancing the understanding of other paleo ice sheets
and in answering key glaciological questions on former ice dynamics (e.g. Dyke, 2004; Bentley et al., 2014; Hughes et al.,
2016; Davies et al., 2020; Dalton et al., 2020; Clark et al., 2022).

Based on motivations outlined above, we present an ice-sheet wide isochrone reconstruction of the Greenland Ice Sheet's
Late-Glacial and early-to-mid Holocene evolution informed by an ice-sheet scale dataset of geomorphological and
geochronological ice extent markers, compiled in an open-access database. This reconstruction, hereafter referred to as
PaleoGrIS 1.0, has a temporal resolution of 500 years, and spans 7.5 thousand years, between ~14 and ~6.5 kyr BP, the
period bracketing most terrestrial landform evidence of ice margin change. This reconstruction represents a first attempt at
combining both geochronological and geomorphological records to estimate the pattern and timing of the ice sheet deglacial
retreat. In this manuscript, we describe our methodology in detail to make the reconstruction transparent and reproducible,
with the hope that improved versions of PaleoGrIS can be produced as more empirical evidence arises. We use our isochrone
reconstruction to conduct a series of ice-sheet wide and regional analyses to quantify the natural variation in retreat rates and
areal extent change, and to address scientific questions on the nature and style of retreat dynamics in Greenland during the
Late-Glacial to mid-Holocene period. We envisage that modellers can use this reconstruction to quantitatively compare
deglacial simulations of the ice sheet with a vast quantity of empirical evidence. For example, the reconstruction (made
available in both shapefile and NetCDF formats) could be used to quantitatively score model simulations ran as part of
ensemble experiments, and/or to narrow down parameter spaces when conducting model sensitivity analyses, by testing the
fit with margin extent and retreat pace (e.g. Patton et al., 2017; Ely et al., 2019; Pittard et al., 2022).





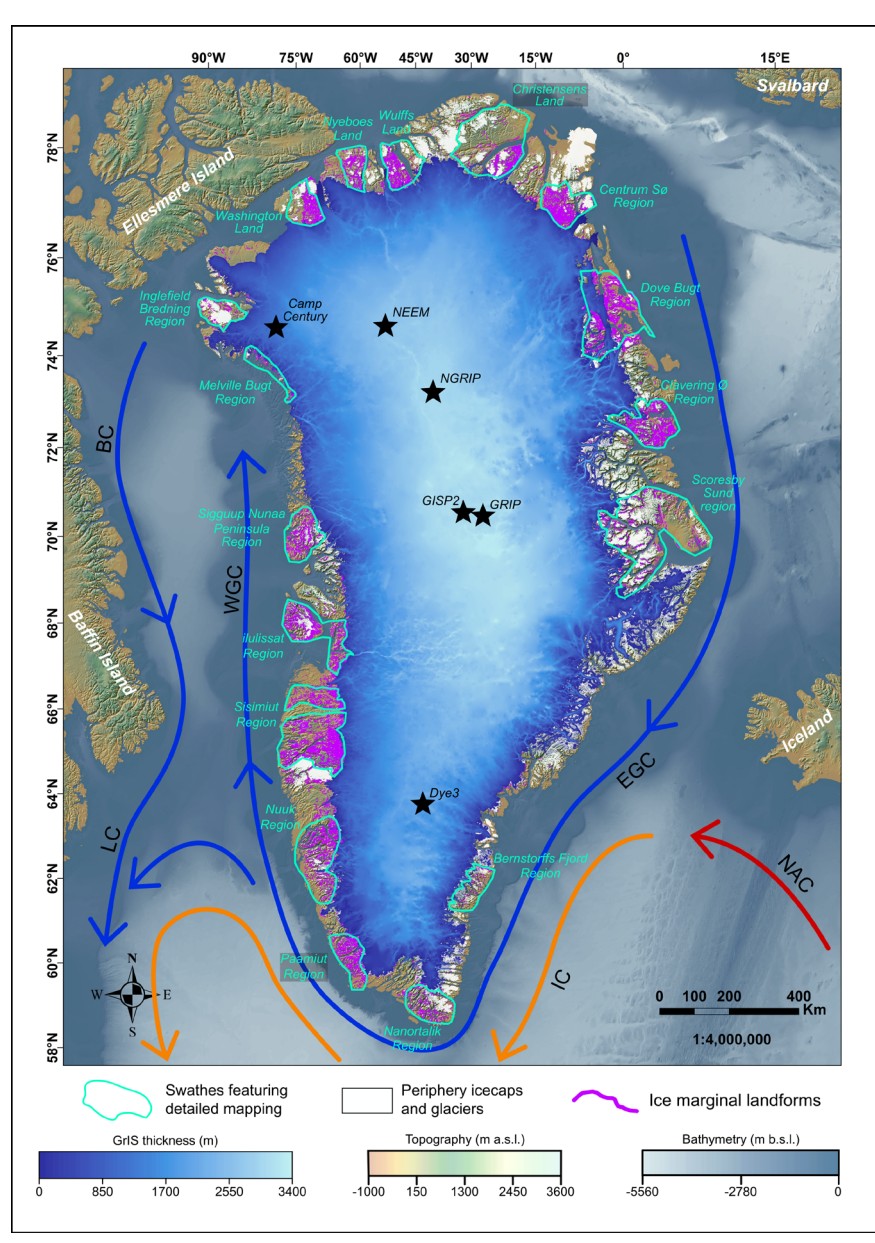

**Figure 1.** Greenland geographical context, present-day ice sheet thickness (BedMachine v4; Morlighem et al., 2017), and location of 18 regions (boxes outlined in light green) where detailed ice-marginal landform mapping (displayed in purple) was conducted as part of this study. Note few areas were mapped in detail in the southeast due to relative lack of ice-free land. A simplified pattern of contemporary ocean circulation is represented, after Yang et al. (2016; Fig. 1); whereby red and orange arrows symbolize warmer Atlantic-origin water while blue arrows represent colder Arctic-origin water. NAC, IC, EGC, WGC, BC, and LC stand for North Atlantic, Irminger, East Greenland, West Greenland, Baffin, and Labrador Currents, respectively. The Digital Elevation Model (DEM) displaying both topographic and bathymetric data is from the GEBCO 2022 release (450 m resolution). Geographical extent of Greenland periphery icecaps and glaciers (white polygons) is from BedMachine v4. The locations of main Greenland ice cores (further mentioned in text) are shown by black stars. All data (here and in subsequent figures) are displayed projected to the WGS 84 / NSIDC Sea Ice Polar Stereographic North coordinate reference system.




## 2 Methods

### 2.1 Geomorphological reconstruction of ice retreat patterns

Across Greenland's exposed land area there has been a rich but sporadic field collection of landform and sedimentary evidence for former ice margin positions. These studies have been based on field investigation and mapping from aerial photographs or satellite images (e.g. Weidick, 1968; 1971; Ten Brink & Weidick, 1974; van Tatenhove et al., 1996; Roberts et al., 2009; Levy et al., 2012; Young et al., 2013), and more recently using Digital Elevation Models (DEM, e.g. Carrivick et al., 2017; Pearce et al., 2018). Such investigations have typically mapped individual moraines, meltwater channels, and 180 trimlines to define former ice margin positions, which were then used to build local reconstructions of ice margin retreat. This information covers a small fraction (<10%) of the land area leaving most of Greenland's ice marginal history unexplored. The release of the 2m ArcticDEM (Porter et al., 2018) for the whole of Greenland provides a consistent dataset that has the potential to revolutionise our understanding of Holocene ice retreat from the Greenland coast to the present-day ice sheet margin position. Motivated by the availability of this new resource, we devised a mapping scheme and protocol that 185 could capture the first-order pattern of retreat for the whole of Greenland's terrestrial ice sheet periphery, an area of 430,500 km$^2$. We did not identify and map every ice marginal landform from the ArcticDEM, as this would represent a decade-long task. Instead, we sampled the area in sufficient detail to provide the first landform-based reconstruction.

### 2.1.1 Geomorphological mapping


We focussed our investigation on landforms indicating the position and shape of former Greenland-Ice-Sheet margins, paying less attention to peripheral ice caps and mountain glaciers. To capture high-resolution details of ice marginal retreat across the whole area and with a consistent approach is a challenge. To accomplish this, we adopted a sampling approach where for some regions, that we called 'swathes', we investigated and mapped ice-marginal landforms in detail (e.g. Fig. 1-195 3). For the intervening areas between swathes, we only identified the most prominent landforms that would permit us to connect paleo ice margins across these areas, joining up swathe to swathe (see section 2.1.2). Eighteen swathes were chosen based on presenting especially dense and well-preserved ice-marginal landforms (Fig. 1), to be positioned approximately evenly around the ice sheet perimeter, and which covered ~60 % of Greenland's ice-free periphery. Although it was beyond the scope of this continental-scale investigation to map all ice marginal landforms identifiable from the ArcticDEM, such an



endeavour would be a valuable future goal. We suggest the main focus should be to pursue mapping efforts in between the 18 swathes, which could contribute to updating future versions of the PaleoGrIS database.

All geomorphological mapping was conducted using remotely-sensed data. Identification of ice marginal landforms was carried out using the 2m ArcticDEM (Porter et al., 2018) and the 30m ALOS WORLD 3D (AW3D30) DEM[1]. As

recommended by Smith & Clark (2005), optimised orthogonal hillshaded relief models of 315° and 45° azimuth angles and 45° inclination were toggled between to minimise azimuth biasing and better identify landforms of interest. Moreover, to aid us in distinguishing glaciogenic sediments from bedrock features, Google Earth Pro software was used for consulting three-dimensional visualisations of satellite imagery (as recommended by Chandler et al., 2018). All mapped landforms were digitised manually as shapefiles in the WGS 84 / NSIDC Sea Ice Polar Stereographic North (EPSG code: 3413) reference

coordinate system. To remain consistent throughout, and to avoid introducing bias from other investigators' landform interpretations, our mapping was conducted without input from previous mapping investigations. Furthermore, mapping was not conducted at a fixed spatial scale, but by zooming in or out to enable better visualization of spatially-variable landform details.

The main indicators of former ice extent were terminal and lateral moraine ridges, more expansive moraine complexes, hummocks and hummocky ridges, lateral meltwater channels and trimlines. These landforms were interpreted and identified based on their morphology and texture, their position in relation to the wider topographic setting, and following criteria for landform interpretation as detailed in Benn & Evans (2014), Chandler et al. (2018), Barr & Clark (2009), Rootes & Clark (2020), and Leger et al. (2020). The most numerous landforms were moraine ridges, which are typically discerned on DEM

hillshades as arcuate, steep-sided, sometimes sinuous ridges with positive relief and often displaying sharp crests (e.g. Fig. 2). Given our aim of covering the entire landmass in a single pass with a small number of investigators, we did not individually map and classify landforms into their respective types (e.g. moraine ridge, meltwater channel). Instead, we captured and summarised information from these landforms by digitising lines all grouped into a single layer (shapefile) called 'ice marginal landforms' (Figs. 2,3). Based on the assumption that glaciogenic deposits relating to previous

glaciations were overridden during the last glaciation (Funder et al., 2011) and unlikely to be preserved, we consider our mapped ice-marginal landforms to have been deposited during the last deglaciation, between ~17 kyr BP and present.

---

[1] AW3D30: https://www.Eorc.jaxa.jp/ALOS/en/aw3d30/



**Figure 2. Comparison between DEM hillshade (light azimuth: 315°, incline: 45°) of the 2 m spatial resolution ArcticDEM (Porter et al., 2018) on the left-hand panel, and the same area with our ice-marginal landform mapping superimposed (right-hand panel). The area presented here is part of J.C. Christensen Land in North Greenland, as highlighted by the red box in figure inset. This region displays remarkable preservation and spatial density of terminal and lateral moraine rides, moraine complexes, lateral meltwater channels and trimlines.**



### 2.1.2 Establishing an internally - consistent map of retreat pattern

The high number (n = 194,302) of ice marginal landforms identified inhibited reconstruction at an ice sheet wide scale. Thus, the landform record underwent two stages of simplification/interpretation, to create a pattern of retreat, following the method employed by Clark et al. (2012). Firstly, 'ice-marginal fragments' were drawn to summarise the collective pattern where ice-marginal landforms could be reasonably linked in close proximity (thick black lines; Fig. 4). As a rule, ice-marginal fragments were mostly confined within the same valley unless the landform evidence was overwhelming for expanding

beyond this scale. A further interpretive, and more speculative, step involved joining up these ice marginal fragments with 'ice-marginal connectors' (thick yellow lines; Fig. 4). These lines attempt to connect nearby ice-marginal fragments interpreted as being approximately time-synchronous. Such connectors are guided by less prominent or less dense spreads of landform evidence and by considering relationships between topography and plausible ice dynamics. As a result, the 18 swathes featuring more detailed mapping (Fig. 1) enable us to connect landforms with both ice-marginal fragments and

connectors, while inter-swathe areas are mostly dominated by ice-marginal connectors. To interpolate ice marginal connectors across offshore areas, submarine topographic data was obtained from the 15 arc-second spatial resolution General Bathymetric Chart of the Oceans (GEBCO) 2022 release[2]. Together, the ice marginal fragments and connectors depict the direction and relative age of ice marginal recession in undated time steps; the first-order pattern of ice marginal retreat (e.g. Fig. 4). The underlying assumption behind our first-order retreat pattern map is that retreat was generally considered to be

monotonic unless there was geomorphological evidence for time-transgressive margins (e.g. cross-cutting moraines).


---

[2] https://www.gebco.net/data_and_products/gridded_bathymetry_data/



**Figure 3. Examples of mapped ice-marginal landforms in two distinct regions displaying high density and preservation of ice-marginal glacial deposits. The mapped landforms are displayed overlaying topographic data from the AW3D30 DEM (30 m resolution). The left-hand panel presents mapping in a region of North Greenland (also referred to as Wulffs Land), while the right-hand panel focuses on the deglaciated region directly South of Disko Bay, in central West Greenland.**

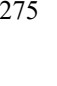









**Figure 4. Visual and cartographic description of methodology followed to produce an ice-sheet-wide retreat pattern from DEM and imagery data. Our procedure involves three main steps of data generalization to incrementally summarise the "raw" topographic and imagery data, to near-continuous 'ice marginal connectors' interpreted as representing time-synchronous former ice sheet margins. The latter product is then used to inform our isochrone time-slice reconstruction, which involves adding geochronological information from our compilation of TCN and radiocarbon-derived event ages.**




## 2.2 Compiling geochronological evidence

We attempt to compile published ages relating to the Greenland Ice Sheet grounded margin retreat from ~14 kyr BP through to the present-day for the entirety of the Greenland domain (Fig 1). We include ages from both Terrestrial Cosmogenic Nuclide (TCN) surface exposure dating and radiocarbon dating methods. Other dating methods (e.g. luminescence,

lichenometry) were not included in this compilation as they have been applied much more sporadically for the specific objective of dating the former position of the ice-sheet margin in Greenland.

### 2.2.1 Terrestrial Cosmogenic Nuclide surface exposure ages

*Compilation of ages*

The geochronological component of the PaleoGrIS 1.0 database compiles published TCN surface exposure ages, and associated metadata, produced with the aim of dating the deglacial and Holocene ice-extent fluctuations of the Greenland Ice Sheet. Such ages estimate the timing of moraine emplacement and stabilisation, deposition of glacial erratics, or the abandonment of ice from exposed bedrock surfaces, all relevant for constraining the timing of ice retreat. These exposure

ages were extracted predominantly from the maintained ICE-D Greenland online database[3]. In addition, a complementary review of the existing literature was conducted to compile relevant studies not currently included in ICE-D Greenland. Exposure ages dating the former margin evolution of periphery ice caps and mountain glaciers were intentionally excluded from our collection. Any new data added to ICE-D and any new study published later than our census date of the 21st October 2022, are not included in the PaleoGrIS 1.0 database.


*Age calibration and re-calculation*

All ages were re-calculated using the online calculator formerly known as the CRONUS-Earth online calculator version 3 (Balco et al., 2008). We do not apply an erosion rate correction to our exposure age calculations. We use the LSDn scaling

scheme (Lifton et al., 2014), and the $^{10}$Be West Greenland production rate (Young et al., 2013) obtained from the ICE-D online calibration database (http://calibration.ice-d.org/). Furthermore, no corrections were applied for post-exposure isostatic and/or tectonic uplift or subsidence, which given the young nature of the ages (<14 kyr BP) and the relatively low magnitude of surface elevation change related to glacial isostatic adjustment during the Holocene, is not thought to cause age offsets greater than analytical uncertainties (Jones et al., 2019). To remain consistent across the domain, no corrections were

applied for post-exposure vegetation or snow cover. Final exposure ages are reported in calendar years (or kyr) before

---

[3] ICE-D Greenland: https://version2.ice-d.org/greenland/publications



present (BP), with 'present' defined as the year of sampling. Nuclide concentrations and sample metadata were retrieved from ICE-D and/or from original publications when needed. When multiple ages were described by original publications as dating a single time-synchronous margin, these were here grouped and summarized by a 'summary event age' (Fig. 5) and uncertainty calculated using the population arithmetic mean and the 1σ standard deviation, respectively. No grouping of ages

was applied if not clearly described in source study as dating the same event. While TCN exposure ages positioned on mountains can act as former ice thickness indicators (e.g. elevation dipstick models) they aren't used directly by our isochrone reconstruction, but are included in the database as they may prove useful for comparing with modelled ice-sheet thickness predictions, for instance.

*Age filtering and quality control*

Stratigraphical and/or statistical outliers were removed from event-age calculations only when considered as such in original publications. The only exception to this is when sample coordinates provided in original publications were either missing or erroneous (e.g. plotting offshore or within the ice sheet). In this case, the age was considered an outlier and was not included in summary event age calculations. For a few more recent investigations (e.g. Søndergaard et al., 2020), in situ cosmogenic

$^{14}C$ was also measured alongside $^{10}Be$ or $^{26}Al$. When $^{14}C$-derived exposure ages display younger exposure ages and are described by authors as presenting no or less nuclide inheritance than other nuclides, they were given priority in our summary event age calculations. Our database also includes the details of whether paired nuclide analysis on a given sample (e.g. $^{26}Al/^{10}Be$) suggests a complex exposure/burial history, and thus guides us to exposure ages that are likely too old to be considered in our Holocene retreat reconstruction. Furthermore, to help combine the geomorphological and geochronological

evidence in a time-slice reconstruction of ice sheet evolution, TCN exposure ages compiled within PaleoGrIS 1.0 are provided a quality control rating and classified in three categories described as high, medium, and low confidence ages. The criteria list followed to apply this quality control is presented in Table 1, and is derived from the investigations of Hughes et al. (2016), Small et al. (2017), and Davies et al. (2020).







**Table 1. Quality control assessment criteria list for TCN and radiocarbon summary event ages compiled in the PaleoGrIS 1.0 database. Criteria are adapted from Hughes et al. (2016), Small et al. (2017), and Davies et al. (2020) to fit the Greenland-specific context of numerical dating of former ice-sheet margin retreat.**

| Dating technique | Quality control rating | Criteria |
|---|---|---|
| Pre-requisites for all techniques | n/a | - All metadata required for age re-calculation/calibration is provided, including AMS standards for TCN exposure ages<br><br>- Publication clearly indicates when multiple ages from a site date an event (i.e. map, table with groups, stratigraphic age model etc)<br><br>- Details of geologic and stratigraphic context is provided<br><br>- Analytical errors are provided and <10% of age |
| TCN exposure | High confidence | - Multiple (at least 3) ages dating an event, post outlier removal<br><br>- Exposure ages dating single event display little scatter: acceptable reduced Chi-square statistic<br><br>- No evidence of complex exposure/burial history<br><br>- No indication of major nuclide inheritance or post-depositional disturbance signals |
| | Medium confidence | - Only 2 samples dating an event, but showing consistent ages (post outlier removal)<br><br>- Only 1 sample but for which the 14C radionuclide was also measured, and it displays a consistent or younger exposure age than obtained with other nuclides<br><br>- Only 1 sample reported to date an event, but located near other samples (< ~10 km) displaying summary event ages consistent with stratigraphic order of events |
| | Low confidence | - Only 1, geographically isolated sample dating an event (post outlier removal)<br><br>- Multiple samples but highly scattered exposure ages<br><br>- Multi-nuclide analysis shows complex exposure/burial history of sample(s)<br><br>- Coordinates reported by author plot in odd location (e.g. at sea or on ice when study site is terrestrial), thus suggesting innacurate geolocation measurement |
| Radiocarbon dating | High confidence | - 1 sample to date an event (post outlier removal) but terrestrial sample and species of macrofossil/microfossil is identified and clearly reported<br><br>- Multiple (at least 2) consistent samples from a single location or stratigraphic sequence (post outlier removal) to date an event, and which include at least 1 terrestrial sample |
| | Medium confidence | - Multiple consistent ages to date an event but all marine radiocarbon ages<br><br>- Dated material is clearly reported<br><br>- 1 terrestrial sample to date an event but dated material is gyttja, peat, humic acid (soil related) |
| | Low confidence | - Only 1 terrestrial but bulk sediment sample to date an event<br><br>- Only 1 marine radiocarbon age to date an event<br><br>- Poor stratigraphic context with respect to studied event<br><br>- Dated material isn't reported or specific enough to attribute a quality control rating<br><br>- Coordinates reported by author plot in odd location (e.g. at sea or on ice when study site is terrestrial), thus suggesting innacurate geolocation measurement |




### 2.2.2 Radiocarbon ages

*Compilation of ages*

The PaleoGrIS 1.0 database also features a collection of Greenland-wide radiocarbon ages and associated metadata which

provide minimum-limiting age estimates of organic deposition in ice-free conditions following ice retreat during the deglacial and Holocene periods. This collection of radiocarbon ages was assembled by consulting former ice-sheet-scale reviews by Bennike & Björck (2002), Dyke (2004), Sinclair et al. (2016), and Dalton et al. (2020). Other more regional reviews, including those by Rinterknecht et al. (2014), Dyke et al. (2014), Larsen et al. (2014), and Young et al. (2021), were also examined. Furthermore, a review of the existing literature was conducted with the aim of finding other relevant studies

not included in the above. Any new radiocarbon date published after the 21st of October 2022, our census date, is not included in the PaleoGrIS 1.0 database.

*Age calibration and re-calculation*

All radiocarbon ages were consistently recalibrated using the IntCal20 curve (Reimer et al., 2020) for terrestrial samples, the

Marine20 (Heaton et al., 2020) curve for marine samples, and the CALIB 8.2 online calibration software[4]. We report final calibrated ages and uncertainty as the midpoint ± half of the calibrated age range at 95% probability (2σ). Calibrated ages are reported in calendar years (or kyr) before present (BP), with 'present' defined as year 1950 AD.

For marine samples, we apply a marine reservoir age correction protocol that attempts to consistently account for both

spatial heterogeneity in the reservoir effect itself, and for variability between *ΔR* calibration sites. To do so, we calibrate all marine ages against the Marine20 curve using regional *ΔR* values obtained from the maintained online Marine Reservoir Correction Database[5] (Reimer & Reimer, 2001). For each sample and location, a final *ΔR* value is obtained by computing the weighted mean of the 10 nearest available ΔR calibration sites, determined directly from the correction database. The reported final uncertainty following this calculation is the maximum of the standard deviation of *ΔR* and the weighted

uncertainty in mean of *ΔR* (Bevington, 1969). Following this protocol, *ΔR* values in our database range from -113 to 73 years, while *ΔR* uncertainties range from 36 to 150 years. We note these *ΔR* values mostly overlap with the newest Greenland-specific marine reservoir age correction assessment of Pearce et al. (*in review*, submitted after our compiled ages were re-calculated). We acknowledge that for polar latitudes (>50ºN), calibrating marine radiocarbon ages against the Marine20 curve may be problematic, due to greater variability in ocean ventilation and air-sea gas exchange caused by

fluctuations in sea ice extent and wind strength, leading to increased and more time-variable marine reservoir effects (Butzin et al., 2005; Heaton et al., 2022). However, this is more likely to be problematic during glacial periods (Reimer et al., 2020).

---

[4] CALIB: http://calib.org/calib/calib.html
[5] http://calib.org/marine/



For polar samples dating to the Holocene (11.5-0 kyr BP), Heaton et al. (2022) recommend calibrating directly against Marine20. Since the PaleoGrIS 1.0 reconstruction spans the Late-glacial and early-to-mid Holocene (14-6.5 kyr BP) period, and because 90% of the calibrated radiocarbon ages compiled in our database are younger than 11 kyr BP, we choose to treat all samples the same, for consistency.

*Age filtering and quality control*

Previous reviews and published studies were systematically filtered so that only radiocarbon dates produced with the aim to constrain the evolution of the ice sheet margin through time were compiled. Following this logic, only dated events that present a clear stratigraphic link to the evolution of the former ice sheet extent are incorporated in the database. In Greenland, a common example of such dates are multiple radiocarbon dates down a core of lacustrine sediments featuring a clear sedimentological transition from subglacial (e.g. till) to proglacial (e.g. silts and clays) deposits. In the latter case, and for any deglacial chronologies presenting multiple radiocarbon ages from a single core or single location, the oldest age of the sequence was considered the closest age estimate of ice-free conditions and organic deposition following ice retreat, and was thus retained as the summary event age (following Hughes et al., 2016; Small et al., 2017; Dalton et al., 2020) (Fig. 5). The data were also filtered so that radiocarbon ages dating the evolution of Greenland periphery ice caps and mountain glaciers were not included in this collection. Stratigraphical and/or statistical outliers were not included in the summary event age compilation when described as such in original publications. Samples featuring missing or erroneous geographical coordinates were also considered as outliers. Radiocarbon ages compiled within PaleoGrIS 1.0 are given a quality control rating and are classified in three categories described as high, medium, and low confidence ages. The criteria list followed to apply this quality control consistently through the dataset are displayed in Table 1.







**Figure 5. Spatial distribution of radiocarbon - derived (left-hand panel) and TCN exposure age - derived (right-hand panel) summary event ages (post filtering and statistics) used to produce the PaleoGrIS 1.0 isochrone reconstruction. Topographic and bathymetric data (GEBCO 2022 release) is overlaid with 1000 m interval contour lines. The dashed line shows our division scheme of the ice-sheet in seven main drainage regions that we refer to throughout the paper and in associated quantitative analyses, after the drainage basin sampling scheme of Rignot & Mouginot (2012), labelled on the right hand panel. The right-hand panel also features lines denoting our estimations of maximum and minimum scenarios of last full glacial Greenland Ice Sheet extent (grounded ice only), based on a review of the literature (see section 2.6), and which informs the mapping of our outermost isochrones.**





**Table 2. List of publications and associated numbers of compiled ages in PaleoGrIS 1.0 for TCN surface exposure dating.**


**TCN exposure ages**

| Publication | Number of ages compiled | Publication | Number of ages compiled |
| --- | --- | --- | --- |
| Andersen et al. (2020) | 13 | Lesnek et al. (2018) | 24 |
| Balter-Kennedy et al. (2021) | 5 | Levy et al. (2012) | 9 |
| Briner et al. (2013) | 2 | Levy et al. (2016) | 16 |
| Carlson et al. (2014) | 29 | Levy et al. (2018) | 16 |
| Ceperley et al. (2020) | 71 | Levy et al. (2020) | 41 |
| Corbett et al. (2011) | 30 | Nelson et al. (2014) | 11 |
| Corbett et al. (2013) | 19 | Philipps et al. (2017) | 6 |
| Corbett et al. (2015) | 28 | Reusche et al. (2018) | 33 |
| Cronauer et al. (2016) | 11 | Rinterknecht et al. (2009) | 12 |
| Dyke et al. (2014) | 23 | Rinterknecht et al. (2014) | 7 |
| Garcia-Oteyza et al. (2022) | 39 | Roberts et al. (2008) | 12 |
| Hakansson et al. (2007) | 4 | Roberts et al. (2009) | 16 |
| Hakansson et al. (2007b) | 7 | Roberts et al. (2013) | 17 |
| Hughes et al. (2012) | 12 | Skov et al. (2020) | 25 |
| Kelley et al. (2012) | 2 | Søndergaard et al. (2019) | 3 |
| Kelley et al. (2013) | 12 | Søndergaard et al. (2020) | 27 |
| Kelley et al. (2015) | 18 | Winsor et al. (2014) | 17 |
| Lane et al. (2014) | 15 | Winsor et al. (2015) | 47 |
| Larsen et al. (2014) | 47 | Young et al. (2013a) | 47 |
| Larsen et al. (2018) | 28 | Young et al. (2013b) | 5 |
| Larsen et al. (2020) | 43 | Young et al. (2020) | 62 |
| Larsen et al. (2021) | 9 | Young et al. (2021) | 61 |
| Larsen et al. (2022) | 47 | | |









**Table 3. List of publications and associated number of compiled ages in PaleoGrIS 1.0 for radiocarbon dating.**

**Radiocarbon ages**

| Publication | Number of ages compiled | Publication | Number of ages compiled |
|---|---|---|---|
| Bennike (1987) | 2 | Kelly and Bennike (1985) | 1 |
| Bennike et al. (1994) | 3 | Kelly and Bennike (1987) | 1 |
| Bennike et al. (1999) | 2 | Kelly and Bennike (1992) | 7 |
| Bennike (2000) | 4 | Kelly and Funder (1974) | 3 |
| Bennike (2002) | 19 | Kelly et al. (1999) | 3 |
| Bennike et al. (2002) | 6 | Kuijpers et al. (2003) | 1 |
| Bennike (2008) | 1 | Landvik et al. (2001) | 3 |
| Bennike and Bjorck (2002) | 7 | Larsen et al. (2014) | 15 |
| Bennike and Wagner (2012) | 1 | Larsen et al. (2021) | 4 |
| Bennike and Weidick (2001) | 75 | Levy et al. (2017) | 7 |
| Bick (1978) | 1 | Lloyd et al. (2005) | 2 |
| Bjork et al. (1994) | 1 | Long and Roberts (2002) | 4 |
| Blake (1987) | 4 | Long and Roberts (2003) | 5 |
| Blake (1992) | 1 | Long et al. (1999) | 12 |
| Blake (1996) | 1 | Long et al. (2003) | 1 |
| Böcher and Bennike (1996) | 1 | Long et al. (2006) | 5 |
| Briner et al. (2010) | 3 | Long et al. (2008) | 2 |
| Briner et al. (2013) | 4 | Manley and Jennings (1996) | 1 |
| Christiansen et al. (2002) | 1 | Marienfeld (1990) | 3 |
| Crane and Griffin (1959) | 1 | McCarthy (2011) | 1 |
| Cremer et al. (2001) | 1 | Nichols (1969) | 1 |
| Davies et al. (2022) | 2 | Ó Cofaigh et al. (2013) | 5 |
| Delibrias et al. (1986) | 1 | Perner et al. (2013) | 4 |
| Donner and Junger (1975) | 3 | Puleo et al. (2022) | 1 |
| Dowdeswell et al. (1994) | 1 | Shotton et al. (1974) | 1 |
| Eisner et al. (1995) | 1 | Simonarson (1981) | 1 |
| England (1985) | 5 | Smith and Licht (2000) | 9 |
| Fredskild (1972) | 2 | Søndergaard et al. (2019) | 1 |
| Fredskild (1973) | 2 | Sparrenbom et al. (2006) | 3 |
| Fredskild (1983) | 7 | Storms et al. (2012) | 6 |
| Fredskild (1985) | 2 | Sugden et al. (1972) | 1 |
| Fredskild (1995) | 1 | Tauber (1966) | 1 |
| Funder (1978) | 2 | Tauber (1968) | 1 |
| Funder (1982) | 7 | Ten Brink and Weidick (1974) | 3 |
| Funder (1990) | 4 | Trautman (1963) | 1 |
| Funder and Abrahamsen (1988) | 2 | Trautman and Willis (1966) | 1 |
| Funder and Hansen (1996) | 2 | Van Tatenhove et al. (1996) | 3 |
| Gulliksen et al. (1991) | 1 | Wagner et al. (2000) | 2 |
| Hakansson (1974) | 1 | Wagner and Melles (2002) | 2 |
| Hakansson (1975) | 1 | Washburn and Stuiver (1962) | 1 |
| Hakansson (1976) | 1 | Weidick (1968) | 1 |
| Hakansson (1987) | 1 | Weidick (1972) | 3 |
| Hansen (2001) | 4 | Weidick (1975) | 4 |
| Hansen et al. (2022) | 2 | Weidick (1976) | 22 |
| Hjort (1979) | 8 | Weidick (1977) | 2 |
| Hjort (1981) | 4 | Weidick (1978) | 6 |
| Hjort (1997) | 2 | Weidick and Bennike (2007) | 10 |
| Ingolfsson et al. (1990) | 5 | Weidick et al. (1990) | 1 |
| Ingolfsson et al. (1994) | 1 | Weidick et al. (1996) | 1 |
| Ives et al. (1964) | 1 | Weidick et al. (2004) | 1 |
| Jennings et al. (2002) | 1 | Willemse (2000) | 1 |
| Jennings et al. (2014) | 1 | Williams (1993) | 6 |
| Kaufman and Williams (1992) | 1 | Williams et al. (1995) | 1 |
| Kelly (1973) | 2 | Young et al. (2011a) | 1 |
| Kelly (1979) | 1 | Young et al. (2011b) | 4 |
| Kelly (1980) | 1 | Young et al. (2013) | 3 |









### 2.2.3 The PaleoGrIS 1.0 geochronological database format

TCN and radiocarbon ages compiled were entered in two respective Excel (.xlsx) spreadsheets, made available in the PaleoGrIS 1.0 database[6]. Both spreadsheets (one for TCN and one for radiocarbon) document sample information and source publication details (also in Tables 2,3), metadata relevant to age calibration and re-calculations, all event identification and summary event ages, and age quality control attributions. A subset of these details including sample names, locations, source publications, summary event ages and uncertainties, were used to generate point shapefiles for use with any geographic information system software (e.g. ArcMap or QGIS). Further details concerning the several shapefiles published alongside this manuscript are described in the *Read Me* files provided in the online database.

### 2.3 Producing Greenland Ice Sheet-wide isochrones

Within the context of reconstructing past ice sheet extent, isochrones are defined as time-stamped and spatially-continuous perimeters highlighting changes in the former spatial extent of an entire ice sheet over time. Reconstructing such continuous perimeters is a challenge given a fragmentary evidence base. Geochronological and geomorphological data available to empirically reconstruct isochrones are either point data (ages) or fragments of line data (e.g. moraine mapping). Such data are characterised by highly variable temporal and spatial densities and are associated with uncertainties of their own. Therefore, the task of drawing time-stamped and spatially-continuous ice sheet perimeters involves interpolating between empirical evidence and extrapolating across blank areas of the map (e.g. Stroeven et al., 2016; Hughes et al., 2016). The following paragraphs describe the methods followed to produce isochrones that combine geomorphological and geochronological evidence while separately accounting for temporal and spatial uncertainties in empirical data.

### 2.3.1 Isochrone time span, temporal resolution, and uncertainty

For PaleoGrIS 1.0, we map isochrones that delineate grounded ice only (and not floating ice fronts) (Fig. 6). Since Greenland ice-free land areas were deglaciated mostly during the Late-Glacial and early-to-mid Holocene periods, there is a higher density of terrestrial TCN and radiocarbon dates displaying ages of between 12 and 6.5 kyr BP (Fig. 7). The number

---

[6] Data available under this link:



of available dates drop significantly between 6.5 kyr BP and 2-1 kyr BP, when the ice sheet was responding to the Holocene
Thermal Maximum (Fig. 7). At this time, the ice sheet is thought to have either been as extensive as today, or more retreated
around most of its perimeter (Larsen et al., 2015; Briner et al., 2016). Consequently, the time period featuring enough
terrestrial geochronological evidence for reconstructing past retreat at the ice sheet scale currently spans only 6-7 kyr. Recent
methodological and technological improvements to TCN exposure and radiocarbon dating now enable the production of
Holocene ages with analytical uncertainties that can be less than to 500 and 200 years (at 1σ), respectively. For these
reasons, we chose to produce ice-sheet-wide isochrones at 500-year temporal resolution between 12 and 6.5 kyr BP.

Between 14 and 12 kyr BP, when it is thought the ice sheet margin was predominantly marine terminating, and when less
data is currently available, we chose to draw isochrones at 1000-year temporal resolution. Prior to this, between the ice
sheet's last full glacial extent and 14 kyr BP, the ice-sheet is thought to have been mostly grounded on the presently-
submerged continental shelf (Dalton et al., 2020). A few offshore sampling studies provide geochronological constraints for
the approximate location of the grounded ice margin during this time period (e.g. Smith & Licht, 2000; Kuijpers et al., 2003;
Ó Cofaigh et al., 2013). However, we believe such studies are currently too scarce and spatially scattered to enable tracing
ice-sheet-wide isochrones between the last full glacial extent and 14 kyr BP, and hope such improvements can be made with
future versions of this reconstruction, as more data from understudied regions arise. The PaleoGrIS 1.0 reconstruction
therefore features 14 isochrones (or time slices) between 14 and 6.5 kyr BP.

In an attempt to account for uncertainties inherent to TCN exposure and radiocarbon dating, we chose to allocate a time
range to individual isochrones, and thus to each time slice of our reconstruction (Fig. 6). For instance, the youngest and
innermost isochrone of PaleoGrIS 1.0 is here referred to as the '7-6.5 kyr BP isochrone.' This means that we estimate the
former margin to have been near the location of that isochrone line at any time between ~7 and ~6.5 kyr BP. This line was
thus drawn to connect, as much as possible, the landforms and summary event ages comprised between 7.0 and 6.5 kyr BP,
when rounded to the nearest 100 years. This approach is different from previous time-slice reconstructions that attributed a
single timestamp to isochrones, and choosing to represent the dating uncertainty spatially by differentiating minimum,
maximum, and/or optimum positions for isochrones for a given time slice (e.g. Dyke & Prest., 1987; Hughes et al., 2016;
Davies et al., 2020; Dalton et al., 2020; Clark et al., 2022). Contrastingly, the PaleoGrIS 1.0 approach aims to more clearly
distinguish and separate the temporal from the spatial uncertainties inherent to isochrone reconstructions. Isochrone temporal
uncertainty is exclusively associated with the analytical and calculation/calibration uncertainties of numerical dates (TCN or
radiocarbon), while isochrone spatial uncertainty instead results from spatially variable density of geochronological and
geomorphological evidence. Here, the latter is treated by attributing various confidence levels along a single isochrone line
(more details in section 2.3.5). Therefore, our approach allows comparing numerical ice-sheet model outputs to a single
isochrone, while enabling model time to vary within our isochrone temporal error range, and thus account for analytical
uncertainty in geochronological data. To help modellers use the PaleoGrIS 1.0 isochrones in their model-data comparison





procedures, our online database includes details (*Read Me* files) on how various data formats could be used depending on model resolution and the type of experiment conducted.


### 2.3.2 Rules followed when drawing isochrones

To draw spatially-continuous isochrones as consistently as possible around the ice sheet's periphery, the following workflow
and set of rules were applied:

• Using ArcMap 10.7.1 software, we displayed our landform mapping database accompanied by our synthesised pattern of retreat map.

• All TCN and radiocarbon summary event ages were displayed rounded to 100 years and featuring a traffic-light colour code relating to one of the three confidence level categories (Table 1).

• All information was displayed on a rendition of topography from the ALOS World 3D 30 m spatial resolution DEM, and bathymetry from GEBCO 2022 release.

• The location of contemporary grounded ice was displayed at all times using the raster mask of the IceBridge BedMachine Greenland version 4 dataset (Morlighem et al., 2017).

• Isochrones were interpreted and mapped working clockwise around the full Greenland perimeter, and sequentially
following chronological order (starting from oldest). The process was conducted iteratively and with numerous adjustments as the position of an individual isochrone might depend upon the preceding and succeeding ones. Multiple authors re-interpreted and contributed to the final isochrones to try and ensure a consensus view of possible alternate behaviours.

• When the ice sheet was everywhere more extensive than present, isochrones were drawn around the whole of Greenland.
However, during younger isochrone time-slices (*i.e.* 10-6.5 kyr BP), paleo ice sheet margins were likely similar to present-day positions, or in a more retreated position, for certain regions (Briner et al., 2010; Larsen et al., 2015). In such cases, isochrones were not drawn around the full ice-sheet perimeter, but were instead interrupted where they meet the present-day margin. We do not attempt to map out the extent of ice where retreated inside of the present-day position as the position of the ice margin is largely undefined/unknown, even within regions where it is known to have occurred.




The precise method employed when drawing isochrone lines varied depending on the nature and spatial/temporal density of empirical data available. When both mapped ice marginal landforms and TCN or radiocarbon event ages coincided, the line was drawn through both sets of information. When only reliable event ages were available, the line was drawn to connect them while considering topography, bathymetry, and the spatial configuration of the modern ice sheet margin (e.g. Fig. 6).

On the other hand, when only mapped ice marginal landforms were available, the line was drawn following the landforms and topography/bathymetry only. In the absence of any chronological constraints, the retreat pattern was assumed to be monotonic in nature, *i.e.* producing a decreasing extent that is spatially and temporally consistent. The resulting ice-margin isochrones are thus much more complex than a product of lines joining up geochronological point data (Fig. 6). Rather the PaleoGrIS 1.0 isochrones represent a qualitative reconciliation and interpretation of topographic, geomorphological, and

geochronological data. Such a heuristic approach sometimes relies on soft knowledge of typical interactions between ice sheet margins and topography, including the lobate behaviour of outlet glaciers and the dynamics of ice flow around present-day Greenland. The isochrones therefore stand as an informed interpretation of successive ice margin positions, in places well constrained and in others much less so, and which are likely to require adjustment once further landform or geochronological data become available.









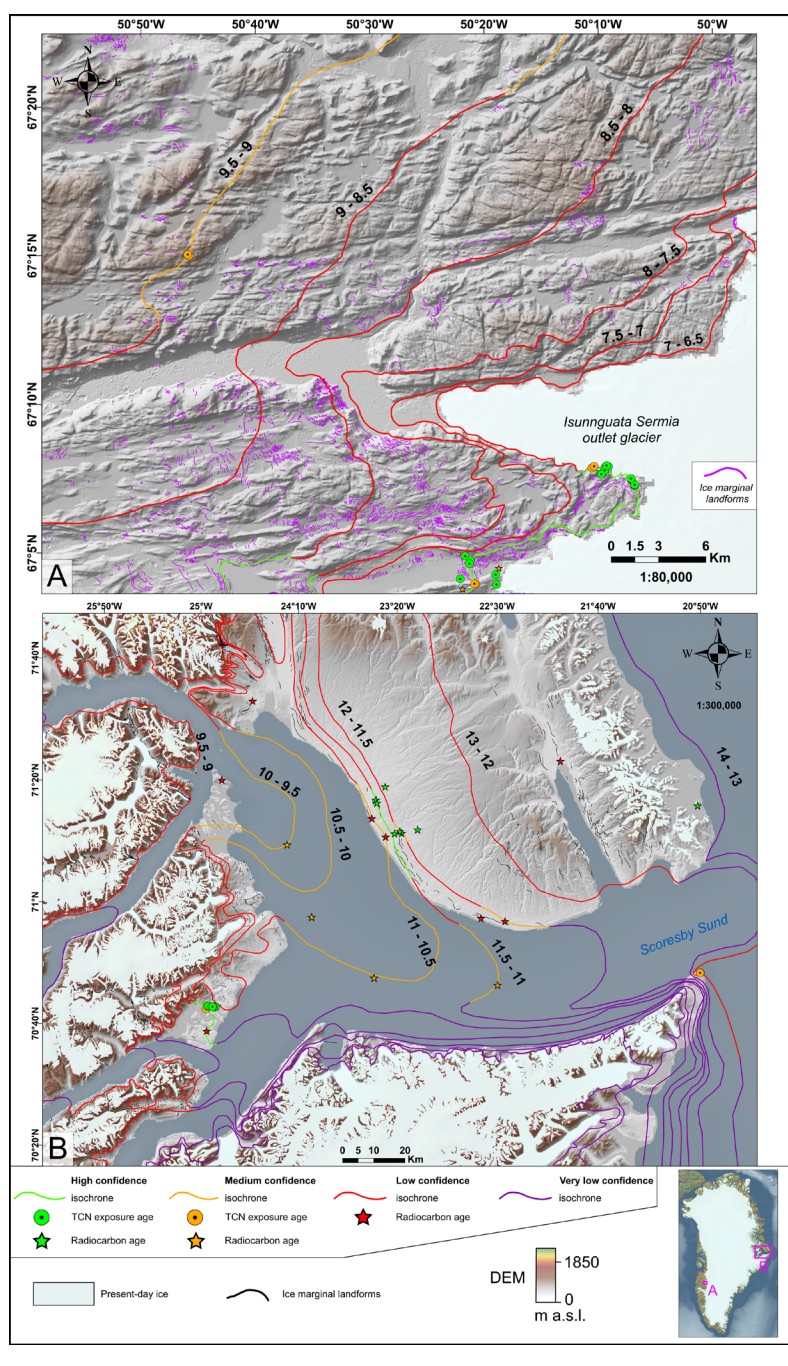

**Figure 6. Example maps showing the details of the PaleoGrIS 1.0 isochrones in a land-dominated region of Central West Greenland (panel A) and a Fjord-dominated region of Central East Greenland (Scoresby Sund, panel B). The two examples highlight our choice of distinguishing isochrone temporal uncertainty (shown by time ranges labelled in bold black) from isochrone spatial uncertainty (shown by the four colour-coded confidence levels). The figure also displays the location of local event ages compiled in our database, here colour-coded based on our quality control assessment (Table 1). All data is displayed overlaying the AW3D30 DEM (30 m resolution).**



### 2.3.3 Addressing periphery ice caps and glaciers

During the Late-glacial and early-to-mid Holocene, as the Greenland Ice Sheet margin receded, peripheral ice caps and mountain glaciers became separated from the main ice-sheet body. Ideally, a full reconstruction of the ice sheet Late-Glacial and Holocene evolution would also include the changing extent of these peripheral ice bodies. Future versions of PaleoGrIS could usefully be extended to include the development of such peripheral ice caps, but for this first version, we chose to exclusively reconstruct the retreat pattern of the main body of the ice sheet. Peripheral ice caps were thus either: i) included
within the perimeter of the reconstructed Greenland Ice Sheet margin, when the latter was considered extensive enough; or ii) excluded from our reconstruction, after complete separation from the ice sheet was estimated to have occurred.

### 2.3.4 Connection with the Innuitian Ice Sheet

Ellesmere Island lies just 100 km to the NW of Greenland (Fig. 1) and along with much of the rest of the Canadian Arctic Archipelago it was covered by the Innuitian Ice Sheet, which is thought to have been connected to the Laurentide Ice Sheet until approximately 10.5-9.5 kyr BP (Dalton et al., 2020). Given such a close proximity it is deemed likely that the Greenland and Innuitian ice sheets coalesced over Nares Strait during glacial maxima (Dyke, 2004; Sinclair et al., 2016; Funder et al., 2011, Georgiadis et al., 2018; Dalton et al., 2020). We chose to make no inferences regarding the pattern of
retreat associated with the margins of the Innuitian Ice Sheet. We thus draw isochrones that are consistent with the two ice sheets merging, but we interrupt them towards the point of deepest bathymetry between Greenland and Ellesmere Island, throughout the length of Nares Strait. The same method was applied to close polygon shapefiles over Nares Strait when measuring the ice sheet's areal extent and create time-slice maps.


### 2.3.5 Isochrone confidence levels

Geochronological and geomorphological evidence are spatio-temporally heterogeneous, causing each isochrone to feature spatially variable levels of reliability around its perimeter (e.g. Fig. 6). Given this uneven distribution of information, our
mapping procedure was adapted accordingly. We split each isochrone line into four confidence-level categories referred to as 'high', 'medium', 'low', or 'very low' confidence isochrones, by applying the following set of rules:

• *High confidence isochrone sections*: Drawn when empirical evidence features both mapped ice marginal landforms and TCN or radiocarbon event ages deemed reliable, *i.e.* graded high or medium confidence.




• *Medium confidence isochrone sections*: Drawn when empirical evidence does not feature mapped ice marginal landforms but features reliable event ages, *i.e.* graded high or medium confidence.

• *Low confidence isochrone sections*: Drawn when empirical evidence features only ice marginal landforms, or only low
reliability dates, *i.e.* ages graded low confidence.

• *Very low confidence isochrone sections*: Drawn in the absence of any mapped geomorphological or any geochronological evidence, but relying on topographic and/or bathymetric information exclusively.


### 2.3.6 Ages that could not reasonably be reconciled to isochrones

When building isochrones by integrating geochronological evidence from numerous published sources and locations,
contradictory evidence can become challenging to resolve. For instance, this can occur when nearby event ages display high age variability while presenting similar levels of reliability, but also when they are found reversed relative to the general direction of presumed ice retreat, and/or relative to stratigraphic order of events in nearby valleys/regions.

For TCN exposure dating, misleading ages can result from nuclide inheritance causing too-old apparent ages, post-
depositional disturbance causing too-young apparent ages, or laboratory contamination/errors potentially causing both (Dunai, 2010). In Greenland more specifically, Late-Glacial and Holocene studies generally report nuclide inheritance to be the dominating cause of exposure age scatter (Larsen et al., 2021). For radiocarbon ages, delayed organic growth following ice retreat, or contamination by younger organics (e.g. for bulk samples) can cause ages to be too young. This challenge is common in the Arctic, where post-deglaciation biodiversity establishment is relatively slow. For marine ages, an erroneous
estimation of the marine reservoir effect can also cause calibrated ages to misrepresent the true deposition age. Moreover, in Greenland, numerous radiocarbon ages were produced to date the marine-to-freshwater transition of basins following isostatic uplift caused by ice retreat (Weidick et al., 2004). Delay in such uplift can increase the potential for radiocarbon ages underestimating the timing of deglaciation. The misfit between our isochrone reconstruction and a specific event age may also be due to oversights in our age-filtering exercise.

In these challenging cases, drawing isochrones requires subjective decisions to either favour and/or ignore certain seemingly awkward or anomalous event ages, when weighed against our interpretation of the most representative timing of local deglaciation. Such interpretations were either based on a review of the region-specific literature, or on assessing the distance separating relevant isochrones in adjacent regions displaying more consistent geochronological constraints. Summary event





ages deemed challenging and ignored were identified and gathered post-mapping of isochrones in a separate shapefile available in the supplement. We acknowledge decisions to ignore event ages are necessarily subjective (see section 2.3.2).

### 2.3.7 Limitations and uncertainties in isochrone reconstruction


In regions of Greenland with high quality and density of geochronological and geomorphological constraints (e.g. the Sisimiut and Disko Bay regions), former ice margins can be reconstructed with reasonably high levels of confidence. Contrastingly, our reconstruction also features crude interpolations of ice margins over vast areas, due to the low density or absence of empirical evidence in numerous locations. Thus, the heterogeneous nature of our reconstruction's uncertainties,

depicted by our four different confidence levels (section 2.3.5.), calls for caution. One must stress that the PaleoGrIS 1.0 isochrone reconstruction is not intended for use at the valley or regional scale, for which local investigations are likely more accurate (e.g. Pearce et al., 2018), but rather at continental scale. Caution should also be applied concerning the temporal resolution of our reconstruction, which aims to provide estimates of ice sheet margin positions every 1000-500 years. In fact, our reconstruction should be regarded as averaged or net retreat over such time intervals, and any short-lived re-advances or

dynamic ice-sheet margin response (e.g. 10s to 100s years) are not captured in our reconstruction. In most locations, the PaleoGrIS 1.0 isochrones should not be considered precise enough to accurately depict the margin evolution of individual outlet glaciers, to predict the formation of small proglacial lakes and spillways, nor for reconstructing past meltwater pathways, for instance. The reliability of the reconstruction is of course dependent on the underpinning evidence, examination of which should guide a user away from more speculative areas. The continental-scale mapping approach can

sometimes cause resulting ice configurations to be glaciologically unrealistic and inconsistent with a complex landscape at the valley scale (< 5 km spatial resolution), despite our efforts to consider the influence of local topography/bathymetry as much as possible.

Over the past 30 years, several offshore investigations have established the approximate location of the ice margin during

initial deglaciation from its local full glacial position (~17-12 kyr BP), when the ice sheet was mostly marine-terminating (e.g; Smith & Licht, 2000; Nørgaard-Pedersen et al., 2008; Evans et al., 2009; Ó Cofaigh et al., 2004; 2013; Rasmussen et al., 2022; Hansen et al., 2022; Lloyd et al., 2023). Such studies remain scarce, and we find former grounding line positions remain largely understudied and undated. Readers should thus be aware that our oldest isochrones (*i.e.* between ~14 and ~12 kyr BP) are associated with crude interpolations across large offshore areas, thus presenting higher levels of uncertainty than

we now have for the terrestrial areas.





We believe there is much scope for reducing isochrone uncertainty in future versions of the PaleoGrIS reconstruction, as more mapping and dating is conducted from less studied regions. We make suggestions as to what regions would most benefit from this attention in a later discussion section.


## 2.4 Estimating areal extent change of the Greenland Ice Sheet

To evaluate the former areal extent of the ice sheet (*i.e.* two-dimensional surface area) between ~14 and ~6.5 kyr BP, we produced a series of polygon shapefiles covering the area delineated by each isochrone perimeter and measured their areal extent using the ArcMap geometry calculator. To compare variations regionally, we divided the ice sheet into seven major hydrological basins, following the ice-divide sampling of Rignot & Mouginot (2012) and the IMBIE Team (2019) (Fig. 5). Region-dividing polylines were extended towards the outermost isochrones following topographic/bathymetric highs, while remaining as perpendicular to isochrones as possible. Therefore, for the exclusive purpose of subdividing the ice sheet into regions, the catchment areas were crudely assumed to remain similar from ~14 kyr BP onwards, as their paleo configurations remain unknown. Between the 14-13 and 10.5-10 kyr BP time slices, we consider our measurements as absolute estimates of areal extent. However, for time slices between 10-9.5 and 7-6.5 kyr BP, our measurements should be regarded as maximum-limiting estimates of ice-sheet areal extent. That is because empirical evidence suggests the ice sheet was as extensive or smaller than today in several regions during that time interval. In these cases, we chose not to guess the extent of retreat behind the current margin, and merged polygons representing more extensive margins with the contemporary ice sheet extent in data-free regions. Present-day ice sheet areal extent was computed from BedMachine Greenland version 4 (Morlighem et al., 2017), after removing peripheral glaciers.

We chose not to convert our ice sheet areal extent reconstruction into a volume and mass estimation. The Greenland Ice Sheet was characterised by very cold and arid conditions during extensive advance, and conversely significant increases in accumulation occurred during deglacial margin retreat. The relationship between areal extent and ice thickness is thus complex and not necessarily positively correlated across the ice sheet (Cuffey & Clow, 1997). To obtain a realistic deglacial volume reconstruction of the Greenland Ice Sheet requires extensive modelling experiments that take into account climate, surface mass balance, glacial isostatic adjustment, relative sea level change, calving, basal sliding (e.g. Bradley et al., 2018), and that conducts quantitative model-data comparisons. Our team is conducting such an experiment with the Parallel Ice Sheet Model (Winkelmann et al., 2011) and will thus present volume reconstructions in a separate publication.





## 2.5 Assessing rates of retreat and their variation


To quantitatively estimate the former retreat rates of outlet glaciers, 72 transects emanating from present-day outlets were drawn. Transects were traced from the outermost (14-13 kyr BP) to the innermost (7-6.5 kyr BP) PaleoGrIS 1.0 isochrone margins, following our estimate of the former glacier front central position at each time step, guided by topography. We thus follow the assumption that the approximate centre of former glacier termini (or grounding lines) was located near the point of lowest topography. Hence, such transects should not be interpreted as supraglacial flowlines, as these would have likely evolved through time with glacier catchments and ice divides potentially re-adjusting during deglaciation. The transects are made available in a polyline shapefile (see online database). Present-day outlet glacier names (preferably new Greenlandic, otherwise Danish) were obtained from the database of Bjørk et al. (2015), while distances along transects were measured for each isochrone spacing using ArcMap. As each isochrone is associated with a temporal uncertainty (*i.e* 1000 or 500 yrs), both minimum and maximum retreat rates were calculated for each time-step. The reported retreat rate for each step is taken as the midpoint ± half the range. For the purpose of evaluating former retreat rates only, the positions of isochrones are regarded as definite, and the reported error is exclusively temporal. For each transect, an 'overall retreat rate' (± temporal uncertainty) is also computed using the total distance divided by the minimum and maximum time span between outermost and innermost isochrones.




## 2.6 Where and when was Greenland's last full glacial extent?


Although we mostly focus on the terrestrial deglaciation from ~14 kyr BP onwards, it was important for positioning this outermost isochrone to have some knowledge of greater ice extents and indeed the maximum achieved extent in the last glacial. A review of Greenland's potential last full glacial extent (thought to occur around 18-16 kyr BP: Simpson et al., 2009) was conducted. We updated the full glacial margin drawn by Funder et al. (2011) after consulting publications contributing new empirical knowledge to that specific question (Möller et al., 2010; Ó Cofaigh et al., 2013; Arndt et al., 2017; Laberg et al., 2017; Jennings et al., 2017; Newton et al., 2017; Seidenkrantz et al., 2019; Sbarra et al., 2022; Couette et al., 2022; Rasmussen et al., 2022; Hansen et al., 2022). In several regions a debate prevails regarding whether grounded ice reached the outer continental shelf when in last full glacial configuration. We find that in all studied regions apart from offshore the Northeast Greenland Ice Stream (NEGIS; Rasmussen et al., 2022), more recent investigations tend to suggest a more extensive full glacial configuration than previously proposed, with grounded ice often argued to have reached the mid-to-outer shelf. For instance, in Central West Greenland, Ó Cofaigh et al. (2013) used a series of dated marine sediment cores and bathymetric subglacial landform mapping to show that the Uummannaq and Jakobshavn Isbræ ice streams remained grounded near the continental shelf edge until ~15 kyr BP. A similar interpretation was, for instance, proposed by Hansen et al. (2022), who argued that grounded ice in the Westwind Trough region (Northeast Greenland) was located towards the







outer shelf during last full glacial extent, and prior to 13.5 kyr BP. However, near the same site (79.5°N), Rasmussen et al.

(2022) suggest that grounded ice did not reach the shelf edge during the last glacial. Such contrasts demonstrate the question

remains open in this region. Vast areas of the Greenland continental shelf are still understudied for this purpose. We revise

the limit of Funder et al. (2011) by considering two scenarios, a minimum and maximum last full glacial extent, with the aim

of highlighting spatial uncertainties in debated and understudied regions (Fig. 5). These two full glacial extent scenarios are

used to inform the mapping of our outermost isochrones, and are included here as shapefiles (see online database). The

maximum extent scenario displays an ice sheet that reaches the shelf edge around the entire ice-sheet perimeter. The

minimum extent scenario modifies the latter by following the outline of Funder et al. (2011) in the Central East and

Northeast regions, except towards the Westwind Trough, where it accounts for new data by Hansen et al. (2022). The

minimum extent scenario also displays a more retreated full glacial ice sheet in the Northwest region, where little data

constrains whether a mid-shelf of outer shelf position was reached. We suppose that the grounding line was likely highly

dynamic and that any maximum achieved last glacial extent configuration between these lines is feasible.






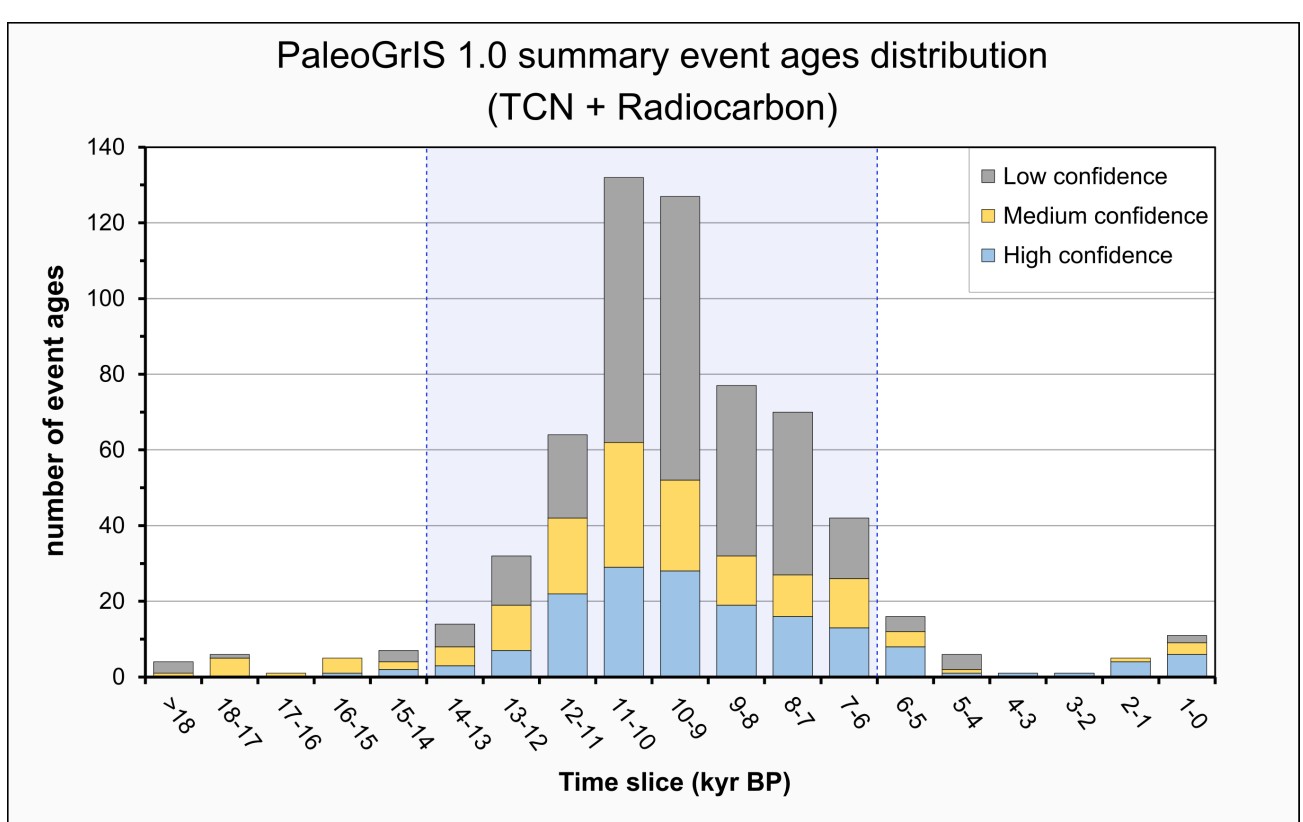

**Figure 7. Histogram displaying the temporal distribution of all summary event ages calculated in the PaleoGrIS 1.0 database. Colour-coded relative proportions of the three quality control categories are shown for each thousand-year time slice. A light-blue polygon highlights the time period covered by the PaleoGrIS 1.0 isochrone reconstruction. This Late-Glacial-to-mid-Holocene period (14-6.5 kyr BP) features the large majority (>90%) of compiled event ages dating the deglacial evolution of former Greenland Ice Sheet margins.**





# 3 Results

## 3.1 The PaleoGrIS 1.0 geomorphological and geochronological database

### 3.1.1 Geomorphology

The PaleoGrIS 1.0 database contains a total of 194,302 ice-marginal landforms mapped over 430,500 km$^2$ of ice-free land around Greenland. The distribution of landforms is relatively sparse in the Southeast (SE) and Northwest (NW) regions, where the contemporary ice sheet terminates near to the coast. Consequently, the majority of ice-marginal landforms mapped

as part of this study are located in the Southwest (SW), Central West (CW), Central East (CE), Northeast (NE) and North (NO) regions of ice-free Greenland. Interestingly, we find ice-marginal landforms are particularly abundant and well-preserved in NO Greenland, increasing our confidence here in the retreat pattern relative to other regions. This abundance and pristine nature of landforms might be due to the generally softer nature of subglacial bedrock (Pedersen et al., 2013) in this region enabling greater sediment supply for moraine-building, a drier climate that could better promote landform

preservation (Niwano et al., 2021), or a potentially steadier and more monotonic retreat pattern of outlet glacier margins characterised by less dynamic fluctuations and re-advances that cause erosion.

### 3.1.2 Geochronology

The PaleoGrIS 1.0 database features 1028 TCN exposure ages gathered from 45 studies and contains 423 radiocarbon ages collected from 111 studies. Following the computation of summary statistics for age groups, and the removal of outliers, this collection contains 251 TCN-derived summary event ages. This number excludes TCN exposure ages produced as past ice-sheet thickness indicators (e.g. dipstick models). The radiocarbon age compilation features 370 summary event ages post age-filtering. Therefore, a total of 621 summary event ages were directly used to inform our isochrone reconstruction.

Details regarding the temporal frequency distribution of summary event ages and their number per investigation are presented in Figure 7 and Tables 2,3. We here acknowledge the possibility that relevant investigations and datasets may be missing from our chronological database. It is our goal that such oversight can be corrected in future versions of PaleoGrIS.

In the PaleoGrIS 1.0 database, 90% (n = 558) of the summary event ages compiled are younger than 14 kyr BP and older

than 6 kyr BP (Fig. 7). As a result, we chose to restrict our isochrone reconstruction to this time period (more justification for this decision can be found in section 2.3.2). Following quality control assessment of summary event ages (Table 1), we find high and medium confidence event ages each represent a quarter of the total (26 and 25%, respectively). Low confidence



event ages represent the largest proportion, *i.e.* 49% (Fig. 7). This is in part explained by numerous event ages resulting from a single marine radiocarbon age, here attributed a low confidence rating due to large uncertainties in past marine radiocarbon
reservoir effects at high latitudes (Heaton et al., 2022) (see criteria list: Table 1). However, we stress that event ages described as "low confidence" may still closely estimate the true timing of margin retreat and are by no means excluded from our isochrone reconstruction. Our event age quality control assessment is merely an indicator utilised to inform challenging decisions made when tracing isochrones.

Using our geochronological database, we analyse the spatial variability in event age reliability across Greenland and its periphery. For TCN exposure ages, we find a higher relative concentration of medium-to-low confidence event ages in NO Greenland, and more specifically from Inglefield Land, Inglefield Bredning, Washington Land and around Danmarks Fjord, Hagen Fjord, Independence Fjords and the Centrumsø regions. Indeed, while studies that investigated these regions sampled both bedrock and erratic surfaces, and have produced extensive datasets, their results often display old apparent ages
(Ceperley et al., 2020; Larsen et al., 2018; 2020; Søndergaard et al., 2019; 2020). It has been stated that this abundance of overestimating exposure ages is most likely associated with high levels of nuclide inheritance in NO Greenland caused by insufficient subglacial erosion of bedrock and transported clasts. Future studies producing exposure ages from these regions might thus be inclined to also measure *in situ* cosmogenic $^{14}$C alongside other radionuclides (e.g. $^{10}$Be, $^{26}$Al, $^{36}$Cl), which may enable the quantification of nuclide inheritance and more accurate estimation of Holocene deglacial exposure ages (e.g.
Søndergaard et al., 2020). We note that the ice-free region to the West of the Akuliarutsip Sermia and Inuppaat Quuat glaciers (67.3-68.3°N; 50.1-54°W), in Central West Greenland, also displays a high concentration of medium-to-low confidence TCN event ages (Young et al., 2020). For radiocarbon ages, however, we do not observe any clear spatial patterns in the variability of summary event age reliability.


## 3.2  Regional retreat patterns and timings

In this section we describe the PaleoGrIS 1.0 isochrone reconstruction, and the pattern and timing of ice-sheet margin retreat, for selected regions, followed by a chronological description of the main deglacial events. This section also describes key
empirical constraints that inform our reconstruction. When found, new Greenlandic place and glacier names are preferably used to describe the geography (e.g. after Bjørk et al., 2015). When not found, we refer to places, features and certain glaciers using Danish or other foreign names.




### 3.2.1 Ice sheet retreat in North Greenland, and the timing of Nares strait opening

Towards the onset of the studied time period (*i.e.* 14-13 kyr BP), our reconstruction portrays the Innuitian (ice over Ellesmere Island) and Greenland Ice Sheets as connected, with grounded ice from both ice masses merging along Nares Strait, as supported by empirical evidence (e.g. Jennings et al., 2011; Georgiadis et al., 2018). For that timestep, we reconstruct the two connected ice sheets to feature marginal grounding lines on both southwestern and northeastern ends of Nares strait, with respective margins terminating into Baffin Bay and the Lincoln Sea (Fig. 8). Geochronological evidence of

ice sheet extent suggests retreat of these two grounding lines occurred somewhat simultaneously on both ends of the Strait. This is supported by ages indicating onshore regions located closest to Nares Strait's openings (e.g. Inglefield Fjord and Wulffs Land) were progressively deglaciated earlier than regions located towards its centre, *i.e.* terrestrial regions adjacent to Kane Basin, such as Washington and Inglefield Lands (Fig. 8). Indeed, TCN exposure ages suggest grounded ice had retreated within Inglefield Fjord by around 11.5-11 kyr BP (Søndergaard et al., 2019). At the same time, deglacial

radiocarbon ages (Funder, 1982; Kelly & Bennike, 1992) indicate ice from the Ryder, Steensby, and C.H. Ostenfeld basins had retreated towards the outermost present-day coastline of Nyeboes, Wulffs, and Nares Lands. After 11 kyr BP, we reconstruct grounding lines that retreated further within Nares Strait, and that reached the southwestern and northeastern edges of Kane basin by 9-8.5 kyr BP (Fig. 8). TCN ages produced from contemporary coastal regions of Inglefield Land (Søndergaard et al., 2020) and Washington Land (Ceperley et al., 2020) suggest regions adjacent to the Humboldt glacier

lateral margins started deglaciating around 8.5-8 kyr BP. During that time, further East, evidence suggests the Petermann Glacier front was located towards the Fjord's mouth (Bennike, 2002). After ~8 kyr BP, in this region, we estimate that the ice sheet had retreated inland beyond present-day coastlines, except for offshore contemporary Humboldt glacier, where we map a grounding line that remained within ~100 km outside the present-day margin. Therefore, empirical data suggests the last ice bridge connecting the Innuitian and Greenland Ice Sheets over Nares Strait survived exclusively within the Kane

basin, and until 8.5-8 kyr BP (although large uncertainties remain). We thus estimate the final deglacial opening of Nares strait to have occurred between 9 and 8 kyr BP (Fig. 8). Following this, further retreat caused the ice sheet margin to reach its present-day extent by 7.5-7 kyr BP on Inglefield Land, while shortly after (7-6.5 kyr BP) on Washington Land.

In northernmost Greenland (>82.7°N), we reconstruct a margin retreat pattern characterised by a disconnection with glaciers

and ice caps from Roosevelts Land that occurred at around 10-9.5 kyr BP. This estimation is however uncertain due to the low abundance of local geochronological constraints. Around 9-8.5 kyr BP, we estimate further retreat to generate ice sheet separation from the large Hans Tausen ice cap. Between 8.5 and 6.5 kyr BP, relatively slow retreat of more-extensive-than present ice margins was still occurring on inter-fjord regions of Wulffs, Adam Berings and Christensens Lands (Larsen et al., 2020). In the same regions, empirical evidence suggests outlet glaciers terminating in deep and wide fjords, such as



Independence, Hagens, or Victoria Fjords, experienced faster retreat than those located on adjacent inter-fjord regions (Larsen et al., 2020).

### 3.2.2    Ice sheet retreat in Northeast Greenland, and ice-margin evolution in the Nioghalvfjerdsfjorden and
**Jøkelbugten regions**

In this section, we describe the PaleoGrIS 1.0 isochrone reconstruction in a region characterised by a ~300 km-long stretch of the coast in northeast Greenland belonging to King Frederick VIII Land, and which lies between Holms Land (79.8°N)
and the Skærfjorden embayment (77.4°N).  The largest contributor to ice flux in this region is the North East Greenland Ice Stream (NEGIS), which currently splits into three wide, marine-terminating and fast-flowing outlet glaciers displaying surface velocities in places > 1000 m yr$^{-1}$ (Joughin et al., 2018). These are, from north to south, the Nioghalvfjerdsbrae (also referred as 79N) glacier, the Zachariae Isstrøm glacier, and the southernmost branch which splits in two sub-outlets, the Kofoed-Hansen Bræ and Storstrømmen outlet glaciers (Fig. 9). In this region, a debate prevails regarding whether the
grounded ice sheet advanced extensively on the wide continental shelf during last full glacial extent, or whether it was restricted to the inner shelf (Rasmussen et al., 2022). However, recent offshore investigations (e.g. Hansen et al., 2022), including soon to be published work (Ó Cofaigh et al., 2023), increasingly suggest a last full glacial ice sheet margin that was more extensive than previously drawn by Funder et al. (2011), and that likely reached the mid-to-outer continental shelf. However, whether the grounded ice sheet remained confined to the prominent bathymetric Westwind (north) and Norske
(south) troughs, or instead also flowed eastwards over the mid-shelf bathymetric highland separating the two, the Northwind Shoal, is uncertain (Arndt et al., 2017; Pados-Dibattista et al., 2022). The large disparity between our minimum and maximum last full glacial extent scenarios reflects this debate in the literature (Fig. 9).

To remain conservative, our reconstruction in this region features a very low confidence outermost isochrone (14-13 kyr BP)
located closely inboard of our minimum last full glacial extent scenario, and depicting a grounded ice-sheet margin positioned towards the mid-shelf, while remaining confined within the Westwind and Norske troughs (Fig. 9). This 14-13 kyr BP extent is constrained in Westwind Trough by a marine radiocarbon deglacial chronology by Hansen et al. (2022). We then tentatively reconstruct an ice sheet margin that had monotonically retreated ~50 km westwards along the troughs by ~12-11.5 kyr BP, as suggested by offshore data by Davies et al. (2022). However, directly south of Norske trough, we
reconstruct a less extensive ice sheet margin around the same time (12-11.5 kyr BP). This interpretation is based on TCN exposure ages by Larsen et al. (2018) indicating that by 11.5 kyr BP, the grounded ice had likely retreated West of Kap Amélie (77.5°N), into the Skærfjorden embayment, and had reached the Storøen and Ambolten Islands; a north-to-south oriented archipelago acting as a topographic barrier to ice flowing eastwards from the deep Jøkelbugten basin (Fig. 9).





Based on TCN-derived event ages from Bourbon Øer (Larsen et al., 2018) and Dove Bay (Larsen et al., 2022) located ~60 km further South, we reconstruct a margin that had retreated to the inner shores of Skærfjorden, and that was positioned ~55 km from the present-day ice front of Nioghalvfjerdsbrae, by ~11.5-11 kyr BP. Over the next two thousand years, our reconstruction suggests the ice sheet margin retreated relatively quickly westwards within the Jøkelbugten basin, through Lamberts Land, and within Nioghalvfjerdsfjorden, with a former margin that was positioned within ~15 km of the present-

day ice sheet front in most locations of the region, by ~10-9.5 kyr BP. We estimate the next phase of retreat to be slower (< 20 m yr$^{-1}$) with an ice-sheet margin remaining more extensive but near the present day one, until ~9-8.5 kyr BP. This is well supported by coeval TCN exposure ages from Bloch Nunatakker, an Island located near Nioghalvfjerdsbrae's contemporary calving front, and from three sites situated ~3 km from the modern lateral margins of Zachariae Isstrøm glacier (Larsen et al., 2018). To the South of Nioghalvfjerdsbrae, and for a 300 km long stretch of the ice margin, we thus reconstruct an ice sheet

margin that was as- or more- retreated than present by ~8.5 kyr BP. This contrasts with the mountainous region directly northwest of Nioghalvfjerdsbrae, however, where the predominantly land-terminating former ice sheet margin appears to have retreated more slowly and more steadily. This is supported by our mapping depicting a highly regular spacing of moraine ridges in this region. There, we reconstruct an ice-sheet margin that remained more extensive than present for two thousand years longer than further South, until ~7-6.5 kyr BP, as indicated by deglacial event ages by Bennike & Weidick

(2001) and Larsen et al. (2018; 2020).











**Figure 8. Time slice maps of the PaleoGrIS 1.0 isochrone reconstruction in Nares strait region. The reconstructed ice sheet areal extent for each given time slice is displayed in each sub-panel as a white transparent polygon, while the underlying opaque white layer is the present-day ice cover from BedMachine v4. The reconstructed ice sheet margins are highlighted by our isochrone polylines, which feature different colour schemes relating to our four isochrone confidence levels (see section 2.3.5). We merge our isochrones with the reconstruction of former ice extent over Ellesmere Island (Innuitian Ice Sheet) from Dalton et al. (2020). While no modifications were applied to the margin extent of that dataset, the timing of Dalton et al. (2020) isochrones was in some cases modified (by less than 1 kyr at most) to match our reconstruction. Bottom panels also feature present-day ice-sheet thickness (Morlighem et al., 2017) and surface velocity data (Joughin et al., 2018). Topography and bathymetry are from the GEBCO 2022 release.**



### 3.2.3 Ice sheet retreat in Central East Greenland, and ice-margin evolution in the Scoresby Sund and Kangerlussuaq regions

The central East (CE) region of the Greenland Ice Sheet is characterised by two major ice-drainage basins. The northernmost one is composed of ice streams and outlet glaciers flowing eastwards, and into Fjord systems that merge to form Scoresby Sund. The southernmost basin comprises three major ice streams flowing south-eastwards, the Kangerlussuaq, Christian IV, and Hutchinson Plateau Glaciers (Rignot & Mouginot, 2012; Bjørk et al., 2015). We here describe the general pattern and timing of retreat in these two key regions (Fig. 10).

*The Scoresby Sund region*

At ~14-13 kyr BP, we reconstruct a grounded ice sheet margin positioned towards the inner continental shelf, near the mouth of Scoresby Sund and around the location of the underwater Kap Brewster moraine complex (Dowdeswell et al., 1994; Fig. 10). The timing of this specific extent is highly uncertain and exclusively based on a TCN-derived chronology from Kap Brewster, by Håkansson et al. (2007). For this time slice, we reconstruct the ice sheet as connected with ice caps from the Liverpool Land peninsula (Fig. 10). However, our reconstruction tentatively suggests disconnection between the two occurred shortly after, towards ~13-12 kyr BP. By ~12-11.5 kyr BP, deglacial radiocarbon ages (Bennike et al., 1999) indicate the Scoresby Sund outlet had retreated within the Fjord, with its northern lateral margin resting against the southwestern slopes of Jameson Land. Following further glacier retreat and thinning, we reconstruct the disconnection of ice flowing south-eastwards within Hall Bredning Fjord and ice flowing north-eastwards within Gaasefjord to occur between ~11.5 and ~10 kyr BP, based on radiocarbon deglacial chronologies by Marienfeld (1990), Ingólfsson et al. (1994) and Hansen (2001). We propose that progressive westward retreat and separation into three outlet glaciers retreating into Nordvest Fjord, Harefjord, and Føhnfjord, occurred between ~9.5 and ~8 kyr BP. After ~8 kyr BP, we estimate the grounded ice extent was similar or less than present in this region (Funder, 1978).

*The Kangerlussuaq glacier region*

During the last full glacial configuration of the ice sheet, outlets of the southernmost basin in CE Greenland converged into one major lobe, commonly referred to as the Kangerlussuaq outlet glacier (Dowdeswell et al., 2010). This major outlet flowed south-eastwards and is thought to have likely reached the outer continental shelf during last full glacial extent (Mienert et al., 1992), although this remains uncertain. The deglacial pattern and timing of retreat of the Kangerlussuaq outlet has been studied by a relatively large number of offshore coring (e.g. Williams, 1993; 1995; Andrews et al., 1996; Smith & Licht, 2000) and onshore (e.g. Dyke et al., 2014) investigations. Based on findings from these empirical studies, we reconstruct a grounded ice margin reaching a mid-shelf position towards ~14-13 kyr BP in this area (Fig. 10). Between ~14



and ~10 kyr BP, our reconstruction suggests the Kangerlussuaq outlet glacier retreated in a relatively slow and monotonic manner (~10 m yr$^{-1}$). Radiocarbon-dated marine sediment records (Smith & Licht, 2000) suggest faster retreat between ~10 and ~9 kyr BP (~70-40 m yr$^{-1}$) caused the outlet glacier to reach the Kangerlussuaq Fjord mouth by ~9 kyr BP. We estimate later retreat within the Fjord to have been relatively quick (~60 m yr$^{-1}$), and a similar or more-retreated than present extent was likely reached before ~ 8 kyr BP (Dyke et al., 2014).

1225

1230

1235

1240

1245





**Figure 9. Time slice maps of the PaleoGrIS 1.0 isochrone reconstruction in the Nioghalvfjerdsfjorden and Jøkelbugten region. The reconstructed ice sheet areal extent for each given time slice is displayed in each sub-panel as a white transparent polygon, while the underlying opaque white layer is the present-day ice cover from BedMachine v4. The reconstructed ice sheet margins are highlighted by our isochrone polylines, which feature different colour schemes (see Fig. 8 for key) relating to our four isochrone confidence levels (see section 2.3.5). Bottom panels also feature present-day ice-sheet thickness (Morlighem et al., 2017) and surface velocity data (Joughin et al., 2018). Topography and bathymetry are from the GEBCO 2022 release.**



### 3.2.4 Ice sheet retreat in the Southeast region, and ice-margin evolution of the Sermilik outlet glacier

To the South of the Kangerlussuaq Fjord and trough, a 300 km-long stretch of the Greenland coastline is characterised by rugged, steep, and high-elevation mountains displaying numerous summits reaching 2000 m a.s.l., with some acting as contemporary Nunataks. These high coastal mountains act as orographic barriers causing snow accumulations and resulting in the contemporary ice sheet reaching the shore along the entire coastline. This results in a lack of settlements, difficult access conditions, and thus a lack of paleo-glaciological field investigations. Moreover, in this region, the offshore

continental shelf does not yet feature published geochronological constraints on deglacial grounded ice margin retreat, as far as our compilation suggests. Therefore, our ice-sheet margin reconstruction along this coastline displays only very-low-confidence isochrones, and retreat is crudely assumed to be monotonic in nature. 300 km further South, the next location presenting empirical constraints on past ice sheet margin evolution is the large Sermilik Fjord, in which the rapidly-flowing Helheim (up to 8000 m yr$^{-1}$: Joughin et al., 2018), Apuseerajik, and Nigertiip Apusiia tidewater glaciers terminate. The

Sermilik Fjord is bordered on its eastern side by the large Ammassalik Island and other peninsulas which feature steep mountains reaching >1000 m a.s.l. These high topographies act as orographic barriers to moisture supply and enable sustaining numerous ice caps and mountain glaciers.

Today, the deep Sermilik Fjord is thus characterised by its main tidewater glacier fronts (e.g. Helheim) terminating more

than ~100 km inside the Fjord, while the Fjord's terrestrial flanks remain heavily glaciated. We attempted to mimic this miscellaneous ice configuration when reconstructing the deglacial retreat of the ice sheet margin in this region. Based on extrapolating chronological constraints from the Kangerlussuaq trough, we map a very low confidence outermost isochrone (14-13 kyr BP) located towards the inner-to-mid continental shelf offshore the Sermilik Fjord. Soon after, by 13-12 kyr BP, we locally reconstruct an ice margin that had quickly retreated towards the mouth of the Sermilik Fjord, and onto the shore

of local coastal mountains. This relatively early retreat scenario is constrained by TCN exposure ages from the southwestern tip of Ammassalik Island (Hughes et al., 2012), located East of Sermilik Fjord, and by exposure ages from the end of the Torqulertivit Imiat valley (Roberts et al., 2008), located near the western edge of the main Fjord mouth. Since deglacial radiocarbon ages from the Ammassalik Fjord further East (Long et al., 2008) give similar ages to TCN exposure dates from rock surfaces located towards the mid-to-inner Sermilik Fjord sides (Hughes et al., 2012), we reconstruct a Sermilik Fjord

outlet front that had retreated ~30 km North by 11.5-11 kyr BP, while ice sheet margins remained extensive, *i.e.* near the coast and towards the Johan Petersen Fjord mouth, on the western side of Sermilik Fjord. The Sermilik outlet thus appears to have experienced a potent state of negative mass balance during the Younger Dryas-to-early Holocene rapid warming transition (GISP2 data; Alley, 2000), while ice masses occupying the Fjord side mountains either remained stable, or retreated less and slower. ~25 km further North, exposure ages from the Amanga Island (Hughes et al., 2012), situated where



the Fjord splits into three, suggest the Sermilik outlet had rapidly retreated further into the inner Fjord, and started to split

into the three main tidewater glaciers existing today (Helheim, Apuseerajik, Nigertiip Apusiia), by approximately 11-10.5

kyr BP. Although uncertain, this rapid retreat prompts us to reconstruct front margins for these three outlets that were as or

more retreated than present after 10-9.5 kyr BP, an early timing for this event relative to other Greenland regions, we find.







**Figure 10. Time slice maps of the PaleoGrIS 1.0 isochrone reconstruction in the Scoresby Sund and Kangerlussuaq region. The reconstructed ice sheet areal extent for each given time slice is displayed in each sub-panel as a white transparent polygon, while the underlying opaque white layer is the present-day ice cover from BedMachine v4. The reconstructed ice sheet margins are highlighted by our isochrone polylines, which feature different colour schemes (see Fig. 8 for key) relating to our four isochrone confidence levels (see section 2.3.5). Bottom panels also feature present-day ice-sheet thickness (Morlighem et al., 2017) and surface velocity data (Joughin et al., 2018). Topography and bathymetry are from the GEBCO 2022 release.**



### 3.2.5 Ice sheet retreat in Southernmost Greenland


The southernmost region of Greenland is characterised by a relatively narrow (<70 km) and shallow (typically <200 m) continental shelf, and features high elevation coastal mountains with several summits reaching 2000 m a.s.l. The present-day ice sheet margin is generally located within 100 km of the outer coast in this region. Compared with other Greenland regions, little ice build-up was required for the ice sheet to reach the continental shelf edge during the last full glacial extent, and this scenario is considered highly likely by previous investigations (e.g. Funder et al., 2011; Andersen et al., 2020; Levy et al., 2020).


The relatively short distance to the continental shelf, combined with high local Bølling-Allerød warming and the proximity to the Irminger Current enabling warm water incursion, are thought to have caused relatively early deglaciation in southernmost Greenland (Levy et al., 2020). In fact, the well-studied N14 lake record, located on the Kitsissut Islands (-45.18°W; 59.98°N), features the oldest radiocarbon deglacial isolation age (~13.6 kyr BP) in Greenland (Bennike et al., 2002; Puleo et al., 2022). Based on these studies, southernmost Greenland is the only region where we map the oldest isochrone (14-13 kyr BP) onshore, towards the present-day outer coastline (Fig. 11). TCN event ages from Lindenows Fjord (Levy et al., 2020) suggest the series of fjords located East of the Julianehåb ice cap started deglaciating at around ~12 kyr BP. Although a lack of evidence makes isochrone mapping in adjacent northern fjords highly uncertain, our reconstruction displays outlet glaciers retreating and reaching present-day configurations earlier than most Greenland regions, by approximately 10 kyr BP. Therefore, the ice sheet margin was as or more retreated than present for most of the Holocene in this region.




To the West of Uummannarsuaq (also known as Cape Farewell), in the region of the Nanortalik and Narsarsuaq settlements, the timing of ice sheet retreat (from Qaqortoq Bay) is better constrained by empirical evidence. The majority of the local ice basin is currently drained by three fast-flowing outlet glaciers (maximum surface velocities >1000 m yr$^{-1}$); the Eqalorutsit Kangilliit Sermiat, Eqalorutsit Killiit Sermiat, and Qooqqup Sermia glaciers (Bjørk et al., 2015; Joughin et al., 2018). TCN exposure ages from Winsor et al., (2015) indicate these outlet glacier fronts had retreated ~20 km into local fjords by ~13-12 kyr BP, while TCN ages from Nelson et al. (2014) suggest retreat over the next two thousand years caused the outlet glacier fronts to be within ~15 km of present-day's before ~10 kyr BP (Fig. 11). This scenario of retreat is also consistent with a deglacial radiocarbon age produced by Weidick (1975) from the Bredefjord (the Narssaq site). To the Northwest, between Qaqortoq Bay and the large piedmont Sioqqap Sermia outlet glacier, our compilation features no geochronological constraints over a ~200 km-long stretch of ice-free coastal land. We have identified this region as a possible target for future field investigations aiming to reconstruct Holocene ice margin fluctuations. Along this region, low-confidence isochrones were thus mapped solely by linking mapped ice marginal landforms and by extrapolating retreat timings from adjacent areas.





Further North, the next site to display chronological constraints on ice sheet retreat is Kuannersooq Fjord (61.98°N), to the East of the Paamiut settlement, with TCN exposure ages from Winsor et al. (2015) dating an ice sheet margin retreating to the Fjord mouth by 12-11.5 kyr BP. Furthermore, in this valley, Carlson et al. (2014) sampled boulder and bedrock surfaces

less than two km from the present-day ice-sheet margin. The resulting TCN exposure ages suggest the ice sheet had retreated towards its present-day extent by ~10 kyr BP in this sector.










**Figure 11. Time slice maps of the PaleoGrIS 1.0 isochrone reconstruction in southernmost Greenland. The reconstructed ice sheet areal extent for each given time slice is displayed in each sub-panel as a white transparent polygon, while the underlying opaque white layer is the present-day ice cover from BedMachine v4. The reconstructed ice sheet margins are highlighted by our isochrone polylines, which feature different colour schemes (see Fig. 8 for key) relating to our four isochrone confidence levels (see section 2.3.5). Bottom panels also feature present-day ice-sheet thickness (Morlighem et al., 2017) and surface velocity data (Joughin et al., 2018). Topography and bathymetry are from the GEBCO 2022 release.**



### 3.2.6 Ice sheet retreat in Southwest Greenland, and ice-margin evolution in the Nuuk region


The ice-free coastal region to the East and South of Nuuk presents a complex topography with high (1500-2000 m a.s.l.) and steep-sided mountains dissected by several sinuous and deep fjords forming an archipelago. Upstream from these fjords, the majority of the ice discharge is captured by four fast-flowing outlet glaciers (maximum surface velocities: 1000-2000 m yr⁻

¹). From South to North: these are the Sermeq, Kangiata Nunaata Sermia, Akullersuup Sermia, and Narsap Sermia glaciers (Bjørk et al., 2015). Due to easier accessibility, several investigations studying the former ice sheet evolution have taken place in this area, and the Late Glacial and early-to-mid Holocene retreat of the ice-sheet's margin is relatively well constrained. Further north, however, to the East of the Attamik settlement, a large ice-free region displaying flatter topographies with fewer overdeepenings features less geomorphological and geochronological constraints on ice margin

retreat.

In the well-studied valleys, numerous TCN and radiocarbon dates from four parallel fjords, and from three distinct investigations (Weidick, 1976 ; Winsor et al., 2015; Larsen et al., 2014) show consistent results. These constraints enable us to map, with good levels of confidence, an ice sheet margin positioned towards the mouths of the Nuup Kangerlua,

Kangerdluarsunguak, and Sermilik fjords, while positioned towards the middle of the Amelarik Fjord, at ~11-10.5 kyr BP (Fig. 12). Older isochrones (14-11 kyr BP) are less well constrained and are here reconstructed following a low-confidence and monotonic retreat pattern from a mid-shelf position at 14-13 kyr BP. Several extensive investigations sampling along the fjords and towards the modern ice margin have shown that after ~11-10.5 kyr BP, local outlets flowing into the Nuuk Fjord system likely experienced very rapid retreat, with their fronts reaching the inner fjords by ~10 kyr BP (Weidick, 1972;

Larsen et al., 2014; Young et al., 2021) (Fig. 12). According to this retreat scenario, local fjords were deglaciated in less than 1000 years, which represents a retreat rate of 207 ± 69 m yr⁻¹ for the Kangiata Nunaata Sermia outlet glacier during that time. Out of the 72 Greenland outlet glaciers sampled for retreat rate analysis, this rate is the 6th highest maximum retreat rate reached over the reconstructed period. Therefore, this rapid retreat of the ice sheet margin in the Nuuk region between ~11 and ~10 kyr BP was likely a high-magnitude event at the ice-sheet scale.


Following this, numerous multi-nuclide TCN exposure ages from near the contemporary margins of the Kangiata Nunaata Sermia, Akullersuup Sermia, and Narsap Sermia glaciers indicate local tidewater glaciers had reached a similar-to-present extent by ~9 kyr BP (Larsen et al., 2014; Young et al., 2021). Further north, towards a plateau (> 600 m a.s.l.) lying between the Nuup Qinngua Fjord and the Saqqap Sermia outlet glacier, younger deglacial radiocarbon ages (~7.5 kyr BP: Levy et al.,

2017) suggest the land-terminating ice sheet margin retreated at a slower pace. Although this hypothesis requires further

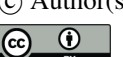



evidence for validation, our reconstruction uses this data to suggest an ice margin remaining more extensive than present until our youngest isochrone, *i.e.* 7-6.5 kyr BP (Fig. 12).











**Figure 12. Time slice maps of the PaleoGrIS 1.0 isochrone reconstruction in the Nuuk region. The reconstructed ice sheet areal extent for each given time slice is displayed in each sub-panel as a white transparent polygon, while the underlying opaque white layer is the present-day ice cover from BedMachine v4. The reconstructed ice sheet margins are highlighted by our isochrone polylines, which feature different colour schemes (see Fig. 8 for key) relating to our four isochrone confidence levels (see section 2.3.5). Bottom panels also feature present-day ice-sheet thickness (Morlighem et al., 2017) and surface velocity data (Joughin et al., 2018). Topography and bathymetry are from the GEBCO 2022 release.**




### 3.2.7 Ice sheet retreat in Central West Greenland, and ice-margin evolution in the Sisimiut, Disko Bay, and the Uummannaq fjord regions


*Sisimiut region*

The coastal region here described as belonging to central West (CW) Greenland stretches 600 km northward from the large Maniitsoq ice cap (66°13′N; 52°11′W) to the Uummannaq Fjord region. From the Maniitsoq ice cap to the southern edge of Disko Bay (Qeqertarsuup tunua), the ice-free coast is characterised by a relatively flat bedrock plateau dissected by deep

fjords and extending ~150 km from the outer coast to the contemporary ice margin (Fig. 13). This region, home to the towns of Sisimiut and Kangerlussuaq, has been the subject of numerous investigations reconstructing the evolution of the ice sheet margin during deglaciation (e.g. Ten Brink & Weidick, 1974; Eisner et al., 1995; Bennike, 2000; Roberts et al., 2009; Rinterknecht et al., 2009; Storms et al., 2012; Winsor et al., 2015; Kelley et al., 2013; 2015; Lesnek & Briner, 2018; Briner et al., 2020; Young et al., 2020). Moreover, ice-marginal landforms are well preserved and highly concentrated in this area.

As a result, the Late glacial and early-to-mid Holocene retreat of the ice-sheet margin is locally well constrained. Based on this high density of geological evidence, we reconstruct with reasonable levels of confidence an ice sheet margin that had retreated to within 30 km (offshore or onshore depending on the valley) of the present-day outer coast by ~12-11.5 kyr BP (Fig. 13). After that time, the eastwards ice-sheet margin retreat appears to have been relatively monotonic in this region. Local outlets glaciers sampled, *i.e.* the South Russel, Isunnguata Sermia, Inuppaat Quuat, and Akuliarutsip Sermia glaciers

(Bjørk et al., 2015), seem to have retreated at intermediate but steady speeds varying between 25 and 30 m yr$^{-1}$ once averaged. By ~9 kyr BP, we reconstruct an ice-sheet margin that was near (within 20 km) the inner end of local fjords (e.g. Kangerlussuaq, Nagssugtoq, Qasigiarssuit). Between ~9 and ~7 kyr BP, empirical evidence suggests the ice sheet started retreating slower and was located less than 30 km from the present-day margin for ~3 kyr in most locations throughout this region. This slowdown in retreat appears to have been coeval with the ice sheet margin becoming fully land-terminating in

most valleys, and is thus a potential consequence of the removal of calving-related ablation. The local reconstructed retreat pattern is such that the ice sheet present-day extent was not reached until ~7-6.5 kyr BP.

*Disko Bay*

Further North, at 68.7°N, lies Disko Bay (Qeqertarsuup tunua), formerly host to a large streaming outlet glacier (also referred as the Jakobshavn Isbræ ice stream) that flowed towards Baffin Bay. This ice stream is thought to have reached the continental shelf break during the last full glacial extent, and likely remained near such extent until ~14 kyr BP (Ó Cofaigh





et al., 2013; Rinterknecht et al., 2014). For our outermost isochrone (14-13 kyr BP), we thus reconstruct an ice sheet margin

located towards the middle-to-outer continental shelf, mostly based on data from Ó Cofaigh et al. (2013) (Fig. 13). Data from McCarthy (2011) and Rinterknecht et al., (2014) enables us to tentatively draw an ice sheet margin that had retreated towards the inner continental shelf, with ice thinning causing its lateral margin to be resting against the southern coastal mountain slopes of Disko Island.

Based on offshore data by Lloyd et al. (2005) and TCN exposure ages by Kelley et al. (2013; 2015), we estimate further retreat into Disko Bay had caused the ice margin to be positioned 30 km offshore the mouth of the Ilulissat Icefjord, and towards the Akunaaq and Qasigiannguit settlements further South, by 10.5-10 kyr BP. The following phase of retreat was likely characterised by local outlet glaciers (*i.e.* Sermeq Kujalleq, Eqip Sermia, Saqqarliup Sermia, Sermeq Avannarleq, Akuliarutsip Sermia) retreating into fjords to the East of Disko Bay. These proglacial regions have been studied extensively,

and are characterised by dense mapping and numerous geochronological constraints (e.g. Donner & Jungne, 1975; Long et al., 1999; 2006; Weidick & Bennike, 2007; Briner et al., 2010; Young et al., 2013; Carlson et al., 2014; Cronauer et al., 2016; Balter-Kennedy et al., 2021). This enables a high density of mid-to-high confidence isochrones to be drawn. To the South of Ilulissat Icefjord, empirical data suggests the ice sheet margin evolution was characterised by slow and steady retreat between ~10 and ~7 kyr BP. At a maximum, local outlet glacier fronts (e.g. Akuliarutsip Sermia) had retreated ~20

km during that time interval, which suggest retreat rates likely below 10 m yr$^{-1}$. Based on these observations, we reconstruct an ice margin that reached present-day extent by ~7.5-6.5 kyr BP in this area. For the Sermeq Kujalleq outlet glacier (also known as Jakobshavn Isbræ), extensive empirical datasets from both sides of the Ilulissat Icefjord enable us to constrain, with high levels of confidence, the position of the outlet's lateral margins through time (Briner et al., 2010; Young, et al., 2013). These data indicate that between ~10 and 8 kyr BP, the glacier calving front was either stable, or slowly retreating,

and remained near the Fjord's mouth with the glacier's right lateral margins terminating on the Ilulissat peninsula (Fig. 13). After ~8 kyr BP, the calving front likely retreated to a mid-Fjord position, with lateral ice margins retreating to within ~5 km of contemporary ones by ~7 kyr BP. We estimate the Sermeq Kujalleq had retreated to near present-day extent between ~7 and ~6.5 kyr BP.


*Uummannaq Fjord*

Located ~250 km north from the former Jakobshavn Isbræ ice stream, a major outlet glacier, known as the Uummannaq ice stream (Lane et al., 2014), also formerly advanced towards Baffin Bay with rapid flow causing formation of streamlined subglacial bedforms (Ó Cofaigh et al., 2013). Radiocarbon ages from a marine sediment core (VC45) suggest that during the

last full glacial configuration, this outlet glacier likely reached the continental shelf break, and remained in this position until ~15-14.5 kyr BP (Ó Cofaigh et al., 2013). Given this maximum-limiting constraint, we draw our outermost isochrone (14-13 kyr BP: very low confidence) in this area towards the middle-to-outer continental shelf, and map the Uummannaq ice stream





margin as a protruding lobe. A radiocarbon-derived event age by Bennike et al. (1994) indicates the ice-sheet margin had retreated from the lower slopes of Hareoen Island prior to ~12.2 kyr BP. We thus tentatively reconstruct South and North
lateral margins of the Uummannaq outlet positioned towards the outer contemporary coast between ~13 and ~12 kyr BP, while mapping a protruding lobe along the central bathymetric trough with a margin reaching a mid-shelf position.

Further East, a TCN-derived event age produced by Roberts et al. (2013) suggest the highlands of Illorsuit Island were becoming ice free by ~11.5-11 kyr BP. For this isochrone, we therefore reconstruct an ice sheet margin that separates into
two distinct glaciers: with a northern glacier retreating northeastwards into Karrat Fjord, and a southern glacier retreating southeastwards into Uummannaq Fjord. Further retreat led to divisions into 12 distinct outlets retreating in a dendritic series of narrow fjords, now host to the calving fronts of several fast-flowing tidewater glaciers, such as Salliarutsip Sermia (72.0°N), Umiammakku Sermiat (71.7°N), Kangilliup Sermia (Rink Isbræ, 71.7°N), or Sermeq Kujallec (Store Gletsjer, 70.4°N). In agreement with TCN-derived event ages by Lane et al. (2014), our reconstruction suggests that by 10-9.5 kyr
BP, the Umiammakku Sermiat and Kangilliup Sermia glacier fronts had retreated towards the Fjord mouths, near the western edge of Karrat Island, ~50-30 km from their present-day fronts. Therefore, in the northern sector of the Uummannaq Fjord system (Karrat area), we estimate that rapid retreat of the ice sheet margin was experienced between ~11.5 and ~9.5 kyr BP.

Informed by data from the Qarajaq Fjord further South (Simonarson, 1981; Roberts et al., 2013), our reconstruction also
suggests a rapid retreat of the ice margin in the southern Uummannaq Fjord system, between ~11 and ~10 kyr BP. However, after ~10-9.5 kyr BP, northern outlet glacier fronts (Karrat area) appear to have remained relatively stable, with fronts staying more extensive than present until at least 5 kyr BP, hence for significantly longer than most other Greenland regions (Lane et al., 2014). This contrasts with the ice sheet margin evolution in the southern Uumannaq sector, for 60 km further South TCN-derived event ages from Philipps et al. (2017) suggest the lateral margins of the Perlerfiup Sermia (71.0°N)
glacier were within 5 km of the present-day margin by 10.5-10 kyr BP. This dataset further indicates that after ~9.5 kyr BP, this outlet was either as or more retreated than the present-day margin. This southern sector is characterised by lower coastal mountains, more gradual slopes, wider fjords and fewer mountain glaciers today than towards the northern Uummannaq Fjord system. This setting may have caused less ice mass contribution from periphery mountain glaciers acting as tributaries, than further North. 70 km further south, our reconstruction suggests the Sermeq Kujallec (Store) glacier was more extensive
than present until ~9-8 kyr BP (Roberts et al., 2013). Overall, empirical data suggest the early-to-mid Holocene margin response of individual outlet glaciers of the Uummannaq Fjord system was complex and heterogenous, and we try to capture this variability in our isochrone reconstruction.








**Figure 13. Time slice maps of the PaleoGrIS 1.0 isochrone reconstruction in Sisimiut and Disko Bay region. The reconstructed ice sheet areal extent for each given time slice is displayed in each subpanel as a white transparent polygon, while the underlying opaque white layer is the present-day ice cover from BedMachine v4. The reconstructed ice sheet margins are highlighted by our isochrone polylines, which feature different colour schemes (see Fig. 8 for key) relating to our four isochrone confidence levels (see section 2.3.5). Bottom panels also feature present-day ice-sheet thickness (Morlighem et al., 2017) and surface velocity data (Joughin et al., 2018). Topography and bathymetry are from the GEBCO 2022 release.**





### 3.2.8 Ice sheet retreat in Northwest Greenland, and ice-margin evolution in the Upernavik and Aappilattup Ikera
region

To the north of the Uummannaq Fjord region, the ~800 km coastline of Greenland that stretches up to Steensby Land (77°N) is commonly described as belonging to the NW Greenland region. This coastline is characterised by little ice-free land as the present-day ice sheet margin reaches the shore along most of the coastline, and this region features relatively low-lying
coastal topographies. In this sector, according to our compilation, few studies constrain the timing and positions of the former Greenland Ice Sheet margin. The exception to this is the region of Upernavik, however, where two studies (Briner et al., 2013; Corbett et al., 2013) provide numerous TCN exposure age - derived time constraints on the deglacial evolution of the Sermeq (Upernavik Isstrøm) tidewater glacier, which drains a major catchment of the NW Greenland Ice Sheet. There, we use the deglacial TCN exposure ages from westernmost Islands located 15 km offshore the Upernavik settlement
(Corbett et al., 2013) to map a retreated outermost isochrone (14-13 kyr BP) located towards the inner continental shelf, approximately 30 km West of the mouth of Aappilattup Ikera (the Upernavik Isfjord). Younger TCN exposure ages and dipstick models obtained from high topographies that lie further East along the Upernavik archipelago seem to indicate a relatively monotonic and slow (~15 m yr$^{-1}$) ice-sheet margin retreat pattern between 14-13 and 11.5-11 kyr BP. We thus reconstruct the Sermeq outlet glacier front to rest towards the outer-to-mid Aappilattup Ikera Fjord by 11.5-11 kyr BP.
Further inland, TCN exposure ages from an Island adjacent to the historical ice margin (Briner et al., 2013) indicate faster retreat likely caused the ice-sheet margin to recede towards the inner Fjord by 10-9.5 kyr BP. Moreover, according to three deglacial radiocarbon ages from lake cores sampled within 5 km of the contemporary ice front (Briner et al., 2013), we reconstruct a local ice-sheet margin that reached a similar to present-day extent shortly after ~9.5-9 kyr BP. Along the coastline located to the North of the Upernavik sector, our reconstruction mostly features very-low confidence isochrones
crudely depicting a retreat pattern assumed to be monotonic, due to a lack of geochronological and land-based geomorphological evidence.









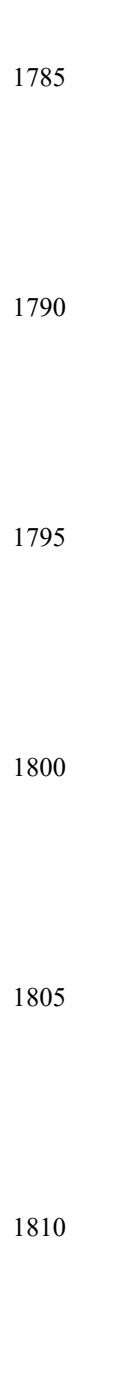

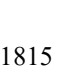





**Figure 14. Ice-sheet-scale map of the PaleoGrIS 1.0 isochrone reconstruction divided in two distinct panels (North Greenland and South Greenland) for readability purposes. The individual isochrone lines mapped as part of this study, and their four respective confidence levels, are too detailed to be visualized at this scale. Instead, we here show a colour map of the areas located between isochrones. Consequently, each colour-coded mask (or 'isochrone buffer') highlights the former location of the Greenland Ice Sheet margins between the timings of the outer and inner isochrones it is delimited by. In order to visualise the full details of the isochrone reconstruction, along with the geomorphological and geochronological data compiled, the reader is advised to download and zoom into the PaleoGrIS 1.0 poster (A0 format) provided in the online database. Alternatively, the reader can download the PaleoGrIS 1.0 shapefile database and visualise the reconstruction at any scale in a GIS software.**

## 3.3 Analysis of changes in areal extent of the Greenland Ice Sheet

### 3.3.1 Ice sheet-wide areal extent evolution

Our reconstruction at 14-13 kyr BP suggests that the Greenland Ice Sheet had an approximate areal extent of 2.61 million km$^2$ (Figs. 14, 15) and lost about one third of this areal extent (0.89 million km$^2$) as it reduced in size to its present-day extent (1.71 million km$^2$; Morlighem et al., 2017). Between 14-13 and 9-8.5 kyr BP, our reconstruction produces a near constant rate of ice sheet areal extent loss of 170 ± 27 km$^2$ per year (Fig. 15). Later, between 9-8.5 and 7-6.5 kyr BP, our reconstruction suggests a rate of areal extent loss that progressively decreased through time. However, as our areal extent estimates are only maximum-limiting between 10-9.5 and 7-6.5 kyr BP (see section 2.4), this potential slow down remains hypothetical and could be a consequence of our limited knowledge regarding how far the ice-sheet margin retreated behind the present-day position in response to the Holocene Thermal Maximum.

At 7-6.5 kyr BP, we estimate the ice sheet areal extent was 1.71 million km$^2$ or less. This suggests the Greenland Ice Sheet extent was similar or smaller than present by this time. Although highly uncertain, our literature-based maximum and minimum full glacial extent scenarios (see section 2.6) suggest the ice sheet areal extent was likely between ~3.13 and ~2.94 million km$^2$ during the last full glacial configuration (Figs. 5, 15). We thus estimate that before the start of the Holocene (~11.7 kyr BP), deglaciation had caused the ice sheet to lose between ~26% and ~21% of its full glacial extent. Therefore, we find that between ~57 and ~51% of the post-glacial areal extent loss occurred before the onset of the Holocene, while a significant proportion (~49-43%) still occurred during the early-to-mid Holocene interval.




**Figure 15. Empirical estimation of the Greenland Ice Sheet areal extent evolution between ~18 kyr BP and present. The ice sheet extent during last full glacial configuration (red box) is here estimated by bracketing maximum (at continental shelf break everywhere) and minimum extent scenarios reconstructed after consulting the relevant literature, in an attempt to highlight the uncertainties in several debated or understudied regions (more details in section 2.6). The hypothetical timing of such full glacial extent is crudely associated with maximum cooling over Greenland during Heinrich Stadial 1 (Buizert et al., 2018), but we stress the timing of maximum extent is unknown and could have been reached before 18 kyr BP, and until BO warming started after ~16 kyr BP, as highlighted by our conservative range (red box). Ice sheet areal extent estimates from the PaleoGrIS 1.0 isochrone reconstruction are shown with green circles (actual extent estimates) and orange triangles (maximum-limiting extent: pointing upwards). This differentiation is due to our isochrones not covering the full ice-sheet perimeter during the ~10-6.5 kyr BP interval, when some sections of the ice sheet margin are estimated to have been as- or more-retreated-than-present margins.**





### 3.3.2 Region-specific areal extent change

After dividing the ice sheet into 7 major drainage basins (after Rignot & Mouginot, 2012; see section 2.4), we analysed
regional patterns of areal extent change through time. While all regions lost areal extent between 14-13 and 7-6.5 kyr BP, we
find the timing, rate, and magnitude of ice-sheet retreat differs substantially between regions (Fig. 16). By 7-6.5 kyr BP, the
NO, SW, and CE regions had lost at least 40% of their 14-13 kyr BP areal extent. The same figure is nearly twice as low
(~25%) for the NW and SE regions. During that time, the CW and NE regions lost at the minimum ~30% and ~35% of their
areal extent, respectively. Furthermore, our reconstruction suggests the rate of areal extent loss, and its evolution through
time, varied significantly between regions. For instance, the SW region lost areal extent relatively quickly from the period
onset (~12-8% kyr$^{-1}$) and reached a similar-to-present extent earlier than other regions, by 10-9.5 kyr BP. The NO region
also lost areal extent at a relatively quick rate (~8-6% kyr$^{-1}$), but such rapid loss started later (after ~12-11.5 kyr BP), and a
similar-to-present areal extent was not reached until ~7-6.5 kyr BP (Fig. 16). In the NE region, on the other hand, empirical
data suggest similar-to-present extent was reached by approximately 8-7.5 kyr BP. Before that time, the rate of percentage
areal extent loss was relatively constant and at ~7-5% kyr$^{-1}$. In the CE region, ice sheet areal extent loss was relatively slow
(~7-5% kyr-1) until ~10-9.5 kyr BP, prior to accelerating substantially (~18-9% kyr$^{-1}$) until ~8.5-8 kyr BP, by which time it
had reached a similar-to-present extent. In our reconstruction, the CE regions thus appears to have experienced rapid and
delayed collapse of a significant proportion of its total areal extent, between ~10 and ~8 kyr BP (Fig. 16). The SE region
reached a similar-to-present extent relatively early, by 9.5-9 kyr BP, and experienced relatively slow areal extent loss prior to
that (~7-5% kyr$^{-1}$). The rate of areal extent loss in the CW region decreased progressively through time, and a similar-to-
present extent was reached late in this region, after ~8.5 kyr BP. The NW region follows a similar pattern, but reached near
present-day extent slightly earlier, prior to ~9 kyr BP (Fig. 16). Such comparisons should be treated with caution, as they are
sometimes derived from low or very low confidence isochrones. However, the data generally suggests a latitudinal signal in
the different regional retreat patterns by which southernmost regions reached a near present-day margin position earlier than
northern regions, which experienced a more delayed deglaciation.







**Figure 16. The PaleoGrIS 1.0 reconstruction of percentage ice-sheet areal extent loss between ~14 kyr BP (0%) and ~6.5 kyr BP, for each of the seven ice sheet regions spatially divided as displayed in inset map, after Rignot & Mouginot (2012). The colour code of individual time series matches the inset map polygons. Importantly, we use a different symbol to highlight whether the percentage areal extent loss, for a given region, is an absolute areal extent estimate (straight line), or a minimum-limiting areal extent loss estimate (dashed line with downward pointing arrows). The switch from straight to dashed lines thus represents, for each region, the youngest isochrone to feature a more-extensive-than-present ice-sheet extent along the full region-specific ice margin.**




## 3.4 Outlet glacier retreat rates analysis

Transects were drawn across reconstructed isochrones for 72 outlet glaciers sampled around Greenland (further methods in section 2.5) (Fig. 17). Measuring distances along these transects allows us to estimate that overall retreat rates of glacier
fronts (or of the grounding line for marine margins), ranged between 72 and 8 m yr$^{-1}$ during the ~14-6.5 kyr BP time period (mean & 1σ S.D.: 37 ± 16 m yr$^{-1}$) (Fig. 17). Although the frequency distribution of overall retreat rates is scattered and bimodal, it is such that >80% of outlet glaciers display retreat rates between 56 and 16 m yr$^{-1}$. Although slightly different in nature, these overall retreat rates are comparable to calving front retreat rates observed by Carr et al. (2017) for marine-terminating glaciers across Greenland for the 1992-2000 AD period (mean & 1σ S.D.: ~42 ± 86 m yr$^{-1}$).


Our reconstructed ice sheet Late-glacial and early-to-mid Holocene retreat rates can be compared with deglacial retreat rates from other ice sheet reconstructions. For instance, Greenland Ice Sheet outlets retreated at paces comparable, or perhaps slightly faster, to terrestrial margins of the British-Irish Ice Sheet, associated with retreat rates < 50 m yr$^{-1}$ for most regions during deglaciation (Clark et al., 2022). Marine calving margins of the British-Irish Ice Sheet, however, generally retreated
faster than our sampled Greenland outlets (>50 and up to 451 m yr$^{-1}$). Greenland outlet retreat rates also appear to be slower than fast-retreating terrestrial margins of the Laurentide Ice Sheet, e.g. the Southwest margin, thought to have reached retreat rates of 380-340 m yr$^{-1}$ following separation from the Cordilleran Ice Sheet (Norris et al., 2022). However, the Labrador sector of the Laurentide Ice Sheet is believed to have experienced slower and similar-to-GrIS-Holocene retreat, with Lowell et al. (2021) reporting a near-constant mean retreat rate of 52 m yr$^{-1}$ between ~19 and ~10 kyr BP in this region. We note that
our reconstructed Greenland outlet retreat rates are net rates averaged over distances and time separating 500-year resolution isochrones. Our reconstruction may thus lead to smoothing of faster pulses of retreat (>200 m yr-1) which therefore should not be ruled out.

Our reconstruction further suggests overall outlet-glacier retreat rates varied significantly between regions (Fig. 17). We find
that outlet glaciers from the CW region were likely the fastest retreating in Greenland between ~14-13 and 7-6.5 kyr BP, with a regional mean retreat rate of ~50 ± 14 (1σ S.D.) m yr$^{-1}$. This region includes the Sermeq Kujalleq (overall rate: 49 ± 5 m yr$^{-1}$), Sermeq Avannarleq (72 ± 14 m yr$^{-1}$) and Store (70 ± 11 m yr$^{-1}$) glaciers, for instance. Outlet glaciers from the NE and NO regions also retreated relatively quickly during that period, with mean retreat rates of 44 ± 10 m yr$^{-1}$ and 44 ± 13 m yr$^{-1}$, respectively. The retreat of outlet glaciers from the CE and SW regions was intermediate in speed, with mean retreat
rates of 35 ± 23 m yr-1 and 29 ± 12 m yr-1, respectively. Finally, we estimate that outlet glaciers from the SE and NW regions experienced slower retreat at the time relative to other regions, with mean retreat rates of 25 ± 5 and 24 ± 15 m yr$^{-1}$, respectively (Fig. 17). The latter contrasts with present-day retreat rates, as marine-terminating outlet glaciers in SE and SW





Greenland currently show the third and fourth fastest regional mean retreat rates (136 and 117 m yr$^{-1}$, respectively) after NO and CW Greenland (Carr et al., 2017). Our reconstruction shows that spatial variability in outlet glacier retreat rate over the

studied period is dominated by these regional patterns. Indeed, we find no statistically significant correlation between individual glacier retreat rates and their present-day front width, nor with the approximate width of over-deepened troughs along which they retreated. Moreover, we argue confidence levels in the ice-sheet-wide PaleoGrIS 1.0 isochrones are often too low to conduct quantitative analyses at the valley or outlet-glacier scale.

Our results also indicate that for most ice sheet outlet glaciers sampled, the rate of retreat is highly variable throughout the reconstructed time period, *i.e.* between 14-13 and 7-6.5 kyr BP. Indeed, the median retreat rate of all sampled glaciers (n = 72) shows intermediate values between ~14 and ~12 kyr BP (~20-10 m yr$^{-1}$), increases between ~11.5 and ~9.5 kyr BP (30-20 m yr$^{-1}$), and decreases to its lowest values between ~9.5 and ~6.5 kyr BP (~15-5 m yr$^{-1}$) (Fig. 18). We thus find a significant proportion of Greenland outlet glaciers experienced acceleration in their retreat pace between ~11.5 and ~9.5 kyr

BP, when our reconstruction suggests numerous glaciers retreated at speeds >100 m yr$^{-1}$ (Fig. 18). These faster retreat rates are not observed as much during other time intervals of the Late-glacial and early-to-mid Holocene. The timing of this faster retreat (~11.5-9.5 kyr BP) coincides with the highest rates of atmospheric and oceanic warming reconstructed between the Younger Dryas (~13-11.7 kyr BP) and the end of the pre-industrial era (Buizert et al., 2018). Furthermore, using linear regression analyses, we find the majority of outlet glaciers (65%) indicate a decelerating retreat pace over the full

reconstructed period (Fig. 18).













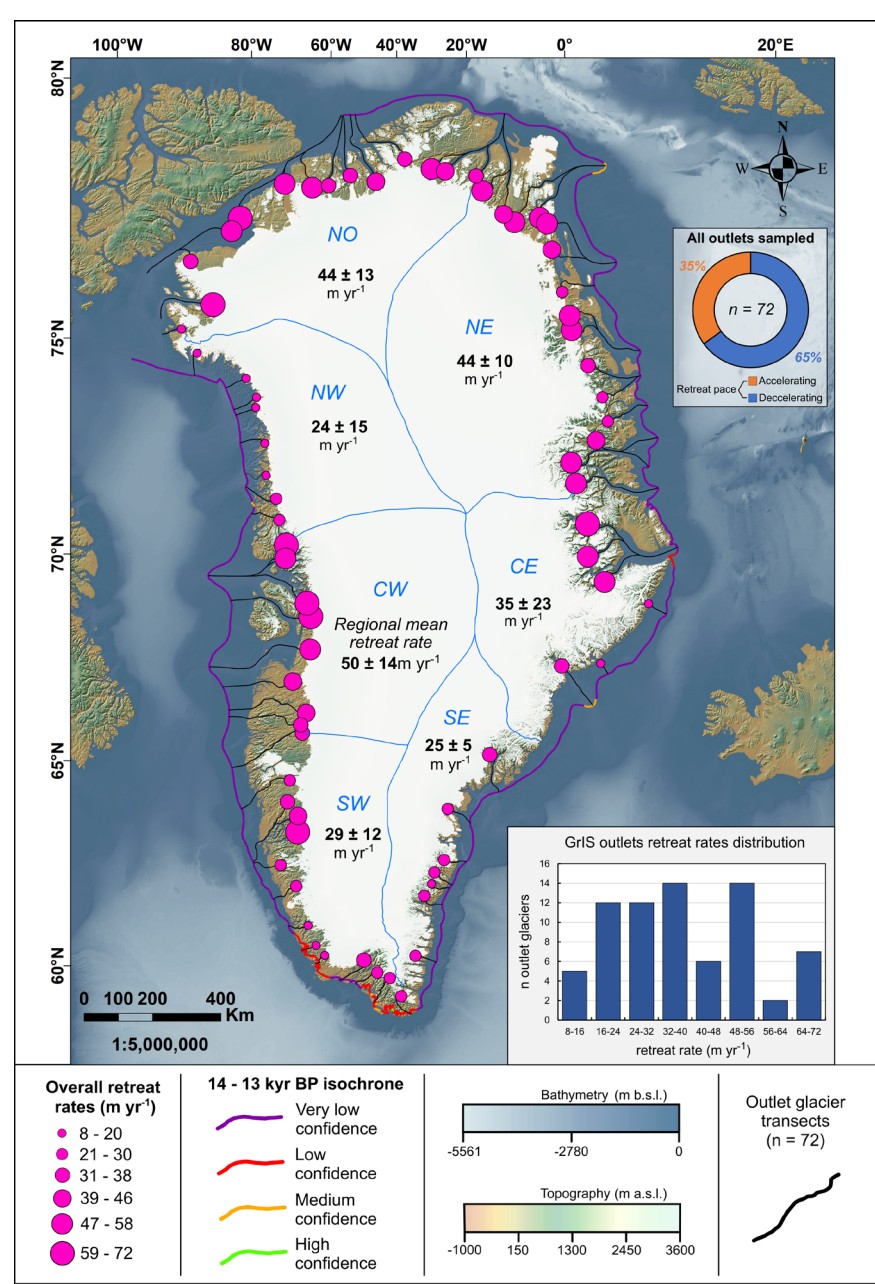

**Figure 17. Map of GrIS outlet glacier transects (black lines, n = 72) drawn to conduct a retreat rate analysis (see section 3.4.) using PaleoGrIS 1.0 isochrones, the main results of which are also shown here. Retreat rates over the full reconstruction period (~14-6.5 kyr BP) are denoted using pink circles characterised by sizes proportional to overall retreat rate magnitudes. Our outermost isochrone (14-13 kyr BP), which marks the outer edge of our transect mapping, is shown with its spatially-variable colour-coded confidence levels. Towards the bottom right-hand corner, a histogram of overall retreat rate distribution is shown as split in eight**
**bins of retreat rate magnitude. Moreover, bold black numbers indicate the regional mean overall retreat rate (± 1σ S.D.) for each of the seven ice-sheet regions sampled. Towards the top right-hand corner, a doughnut diagram highlights the relative proportions of outlet glacier retreat rates indicating decelerating (65%) vs accelerating (35%) trends over the full reconstruction period.**





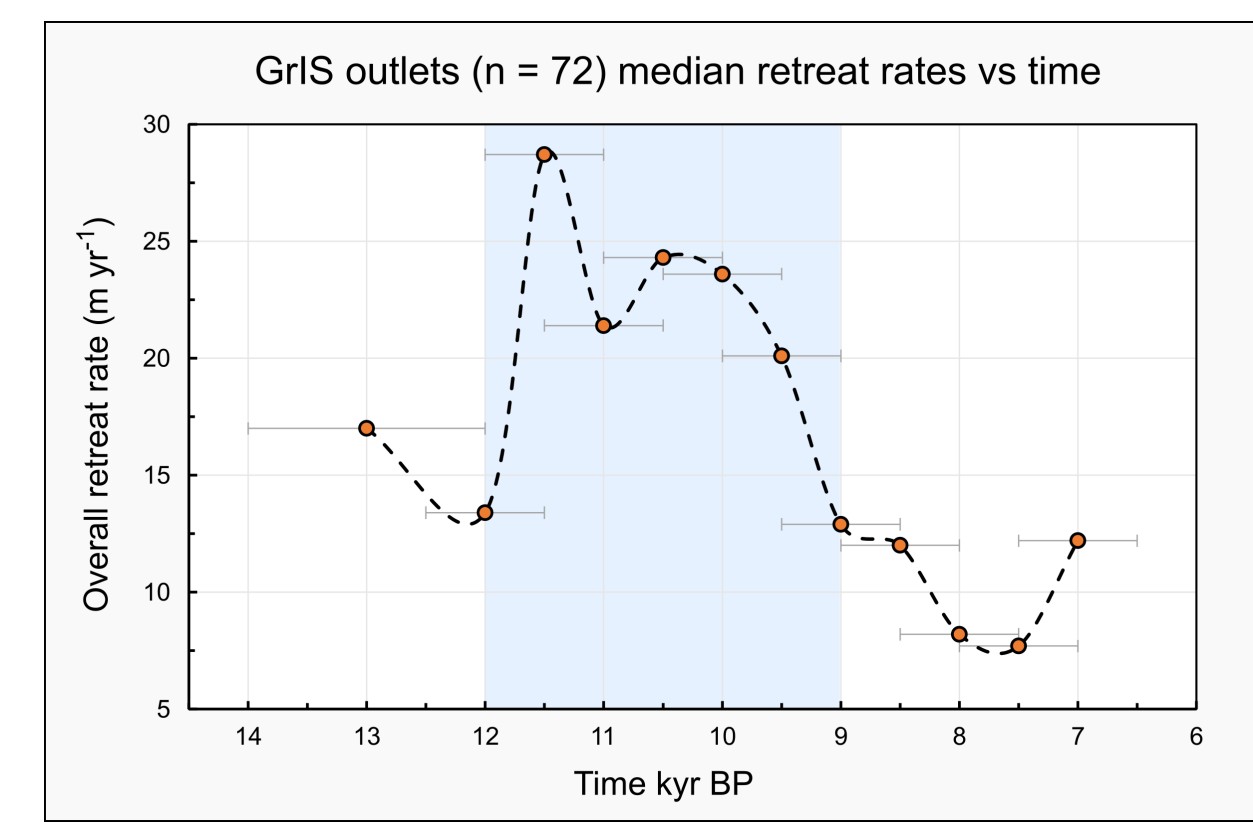

**Figure 18.** Time series denoting the evolution of the median retreat rate for all outlet glaciers sampled over the full reconstruction period (14-6.5 kyr BP). The blue polygon highlights a period of general acceleration in Greenland Ice Sheet outlet glacier retreat speed (see section 3.4). It is important to note that the youngest time slices are here associated with less than 72 data points, as our reconstruction features certain outlet glacier margins reaching present-day extent before others.



## 4. Discussion

### 4.1 Progress towards a robust reconstruction of the Greenland Ice Sheet retreat, knowledge gaps, and opportunities for future investigations


PaleoGrIS 1.0 is the first attempt to merge both geochronological and geomorphological markers of former grounded ice extent to produce an ice-sheet-wide isochrone reconstruction of the Late-glacial and early-to-mid Holocene evolution of the Greenland Ice Sheet margin (Fig. 14). It thus represents a significant improvement from previous ice-sheet-wide reviews and

compilations (e.g. Dyke, 2004; Funder et al., 2011; Lecavalier et al., 2014; Sinclair et al., 2016). Another novelty of this work is that it aims to make the reconstruction and empirical database as accessible and usable as possible, whether it is to inform future empirical and model-data comparison investigations, or to provide a format suitable to the production of future versions of the reconstruction (see online database).

This first version of PaleoGrIS however remains incomplete, and has its limitations. As noted earlier (section 2.1.1), our mapping of ice-marginal landforms was conducted rapidly over a few years and deliberately at a reconnaissance level to identify only the main terrestrial landforms recording former ice margin positions. This was necessary to cover the large area and permitted us to erect the ice-sheet-wide framework we have presented. Conducting and compiling more detailed regional studies that map a wider range of ice marginal landforms, including those located offshore, is an important future task. In

understudied regions (e.g. Fig. 19), our reconstruction may act as a stimulus for such investigations and could represent a template that will undoubtedly require revision. We believe such revision would greatly benefit from becoming a wider community effort enabling to gather newly acquired data and knowledge on the update scenarios of local ice sheet margin history (e.g. DATED-2, Hughes et al., 2023).

To enhance confidence in the empirical record and improve our ability to quantitatively compare paleo model simulations with observations, the glaciology community requires more field data from and around Greenland. We here used our reconstruction to identify regions that would most benefit from new constraints on the timing of former grounded ice-sheet retreat. To do so, we mapped terrestrial and offshore regions displaying no or only low-confidence event ages, and thus presenting either low or very low confidence isochrones in our reconstruction. The highlighted regions are shown in Figure

19, and shapefiles are provided in the online database. Care should be taken as this assessment does not include ongoing research efforts, data yet to be published, or data published after our census date (21/10/2022) (e.g. Weiser et al., 2023). Nonetheless, we find vast terrestrial and offshore areas displaying low or very low confidence isochrones. Offshore of present-day coastlines, we note the majority of the Greenland continental shelf remains data-scarce when compiling studies establishing the timing and pattern of grounded ice retreat from the last full glacial extent. For terrestrial regions, we also





find numerous areas displaying potential for new field data collection, around the full ice sheet perimeter, with a relatively greater density and surface area coverage of less studied sites in the northern half of the island's periphery (>71°N; Fig. 19). It must be noted that such data gaps are most likely related to the increased logistical, safety, access, and financial constraints associated with field investigations in more remote and problematic regions. Furthermore, a great source of uncertainty impeding a better understanding of the deglacial dynamics originates from the ice sheet being generally more extensive

today than between ~6 and ~2 kyr BP, when responding to the Holocene Thermal Maximum. Obtaining empirical evidence of the ice sheet minimum extent during that interval (e.g. as is attempted by the ongoing GreenDrill project: Briner et al., 2021) would thus greatly improve our capacity to reconstruct the ice-sheet deglacial evolution.






**Figure 19. Map of offshore and onshore regions that we interpret as understudied for the specific purpose of mapping and dating the deglacial evolution of the grounded Greenland Ice Sheet margin. We thus believe these regions would most benefit from new geomorphological and geochronological data. They were identified by locating regions dominated by very low or low confidence isochrones in our PaleoGrIS 1.0 reconstruction. Two polygon shapefiles associated with these highlighted regions can be found in the online database.**



## 4.2 Exploring controls on the varied retreat dynamics of the Greenland Ice Sheet

Our overarching interpretation of the reconstruction is that the deglacial retreat pattern and rate of the Greenland Ice Sheet, and the magnitude of the resulting mass loss, are controlled by a series of interacting mechanisms whose relative importance differs depending on the spatial and temporal scales analysed.

### 4.2.1 Ice-sheet-wide response

*Linear response to climate and ocean forcing*

Our reconstruction shows that at a Greenland-wide scale, the ice sheet mostly retreated in a simple radial pattern progressively stepping back from the continental shelf during the last deglaciation (Figs. 14, 15). When analysed at a 1000 to

500-year resolution, the retreat was not found to significantly diverge from a linear retreat pattern other than with local offsets and complexity arising from lagged positioning of marine versus terrestrial margins (e.g. in and adjacent to outlet glacier tongues: Fig. 12) and where the main ice sheet left residual ice caps behind. As we here focus on the ice-sheet wide retreat signal (heterogeneous regional behaviours discussed further in section 4.2.2), and because event ages are now reasonably well distributed around Greenland's periphery, we believe this simple retreat signal would still arise even with

improved spatio-temporal distribution of geochronological markers. Along with this surprisingly straightforward retreat pattern, we observe that the ice-sheet-wide reduction in areal extent reveals a simple (negative) linear relationship with atmospheric or sea surface temperatures (Buizert et al., 2018; Osman et al., 2021) during the Late-glacial and early-to-mid Holocene (Fig. 20). These correlations suggest climate and ocean forcing were the dominating agents of former ice-sheet extent change. For instance, the overall acceleration of outlet-glacier retreat that we observe between ~11.5 and ~9.5 kyr BP

is coeval with the potent atmospheric and oceanic warming that characterised the Younger-Dryas-to-early-Holocene transition in the North Atlantic and Greenland regions (Grootes et al., 1993; Fig. 20). Such observations suggest the Greenland Ice Sheet was highly sensitive to atmospheric and oceanic warming. The findings of a more-retreated-than present ice sheet during and following the Holocene Thermal Maximum (Briner et al., 2014), followed by ice margin readvances during mid-to-late Holocene cooling that peaked with the Little Ice Age (~1850 AD; Kjeldsen et al., 2015), are more

evidence of this broadly linear response to climate and ocean forcing. We suggest that at a whole ice sheet scale, complexities in ice dynamics are less relevant at 1000 to 500-year timescales than is perhaps perceived when analysing contemporary glaciological fluctuations and feedbacks. Alternatively, our reconstructions are not yet sufficient in density and resolution to adequately constrain retreat and advance oscillations that might have existed.






*'Glaciological inertia' and delay in response*

Although the Greenland Ice Sheet was highly sensitive to atmospheric and oceanic warming, specific glaciological responses to a forcing may vary over different timescales. Melting or calving can happen nearly instantaneously whereas changes to ice accumulation may take thousands of years to work through the system. Dynamic changes to flow geometry and the positions of ice divides, as well as responses to glacial isostatic adjustment may take even longer still (Rogozhina et al., 2011). Consequently, significant inertia in the system may have caused delay in the ice sheet extent response to deglacial warming. The PaleoGrIS 1.0 reconstruction suggests that the Greenland Ice Sheet margins remained more extensive than present in several regions until 7-6.5 kyr BP. The minimum Holocene extent of the ice sheet was thus not reached until later, most likely between 6.5 and 4 kyr BP, as was previously suggested by numerical modelling experiments (e.g. Lecavalier et al., 2014). Over Greenland, $\delta^{18}O$ ice core records (e.g. Alley, 2000; Buizert et al., 2018b) and temperature reconstructions from paleo-climate data assimilations (e.g. Buizert et al., 2018a; Erb et al., 2022) all suggest Holocene mean annual temperatures reached maximum values between ~9 and ~6 kyr BP, followed by gradual cooling until the end of the pre-industrial era (~1850 AD) (Fig. 20). Therefore, our ice-sheet-wide empirical reconstruction confirms that during the Holocene, the ice-extent response of the Greenland Ice Sheet lagged the cessation of warming, potentially by thousands of years. Inertia of the Greenland Ice Sheet following warming has been previously suggested (e.g. Yang et al., 2022), and has implications for ongoing climate warming generating committed future mass losses, and thus sea level rise contributions, that may last for centuries to millennia.





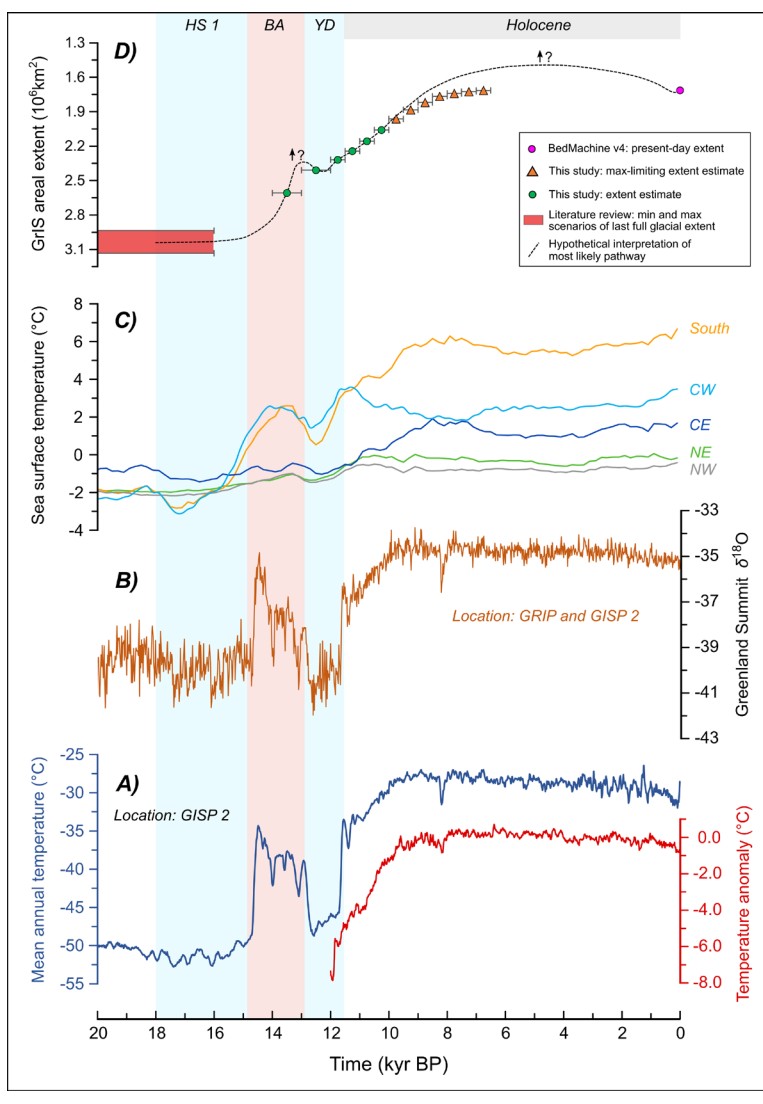

**Figure 20. Comparison between our reconstruction of Greenland Ice Sheet areal extent evolution and past atmospheric / sea surface temperature reconstructions from proxy and model data, between 20 kyr BP and present. A) Blue curve: reconstruction of mean annual near-surface temperature at location of the GISP 2 ice core (72.58°N; -38.48°W) by Buizert et al. (2018a), who merged Greenland ice core temperature reconstructions (GISP2, NGRIP, NEEM) with the TraCE-21ka simulations. Red curve: ensemble mean reconstruction of mean annual temperature anomaly (towards GISP 2 ice core location) relative to a 5-3 kyr BP reference period, by Erb et al. (2022), who use the global Holocene paleo-temperature proxy database of Kaufman et al. (2020) to conduct data assimilation on HadCM3 and TraCE-21ka simulations. B) The Greenland Summit $\delta^{18}$O record (Buizert et al., 2018b) that averages the GISP2 and GRIP $\delta^{18}$O records. C) Reconstructions of sea surface temperature at different locations offshore Greenland, using the ensemble mean (n=500) from the LGMR global dataset of Osman et al. (2021), who use a paleoclimate data assimilation scheme to correct biases in climate simulations of the iCESM (versions 1.2 and 1.3). Coordinates for time series: 'South' (Cape Farewell): 58.34°N; -43.25°W, 'CW' (Davis Strait): 64.72°N; -56.53°W, 'CE' (Denmark Strait): 66.64°N; -27.26°W, 'NE' (Nordic Sea): 72.25°N; -10.10°W, 'NW' (Baffin Bay): 73.17°N; -62.65°W. D) Greenland Ice Sheet areal extent reconstruction from literature review (LGM - HS 1 period), the PaleoGrIS 1.0 isochrones (this study: ~14-6.5 kyr BP), and BedMachine v4 (Morlighem et al., 2017).**





### 4.2.2 Regional responses

Moving from ice-sheet wide to a focus on individual regions (the ice catchments of Rignot & Mouginot, 2012), we find heterogeneous retreat patterns with apparent differences in both the magnitude of regional areal extent loss and the speed of ice margin retreat (Figs. 16, 17). Spatially variable climate and ocean forcings (Fig. 20) may have caused some of these differences. For instance, the saline Irminger Current transporting warm Atlantic waters offshore South Greenland (Fig. 1), likely caused this region to experience early, more intense oceanic and atmospheric warming (Fig. 20c). Such warm water input likely induced faster and earlier deglacial retreat of the then marine-terminating southern ice sheet margins, relative to northern regions (Knutz et al., 2011). However, at the regional to valley scales, we believe non-climatic factors also help explain the variability. We discuss and hypothesise some of these mechanisms in the following sections.

*Influence of heterogeneous continental-shelf topographies*

During Heinrich Stadial 1 (~18-15 kyr BP; Fig. 20), when local atmospheric- and more importantly sea surface-temperatures decreased (Grootes et al., 1993; Osman et al., 2021), Greenland Ice Sheet margins generally advanced across continental shelves (e.g. CW Greenland: Ó Cofaigh et al., 2013). Such advances were facilitated by the soft and unlithified nature of continental shelf surface sediments promoting shear deformation of subglacial sediments, high basal sliding velocities, and ice streaming in some cases (Evans et al., 2009). Certain regions have wider continental shelves providing grounded ice considerable distances across which to advance during build-up phases, while other sectors only feature a narrow shelf to traverse before the continental slope, deep ocean and high calving losses are attained. For example, assuming grounded ice reached the continental shelf break in the NE, offshore of NEGIS, the ice sheet had a distance to advance of ~280-300 km relative to present-day positions. In southernmost Greenland however, this distance to attain the shelf break was instead only ~80-100 km.

During full glacial conditions, precipitation and ice accumulation were very low in Greenland due to cold atmospheric and oceanic temperatures (Buizert et al., 2018). The GISP2 ice core record (Cuffey & Clow, 1997) suggests accumulations <10 cm yr$^{-1}$ between 20 and 15 kyr BP, and the latest simulations of the water isotope - enabled Community Earth System Model (iCESM) suggest values of around 5 cm yr$^{-1}$ for that location (He et al., 2021). Therefore, in sectors experiencing larger lateral ice-sheet expansion across wider continental shelves (e.g. NE and CW regions), ice accumulation was less likely to be able to compensate the required mass transfer to lower elevations during extensive margin advance. This potentially caused greater ice-sheet thinning and lower surface slopes than in other sectors experiencing less expansion across the continental shelf (e.g. SW, NW and CE regions).





Relative to a steeper ice-sheet profile, a thinner sector of the ice sheet with a gentler surface slope should experience increased melting for a given rise in Equilibrium Line Altitude because a greater surface area is now below that altitude 2280 (Edwards et al., 2014). Warmings such as during the Bølling-Allerød and early Holocene intervals may therefore have had varied effects on mass balance arising from regional differences in ice surface slope attained during build-up across continental shelves of different widths. Hypothetically, such mechanisms may have caused Greenland sectors with wider continental shelves and more extensive last full glacial advances (e.g. offshore CW Greenland) to have experienced faster grounding line retreat during the initial phase of deglaciation, under still predominantly marine-terminating conditions (e.g. 2285 Fig. 21b, scenario 2). Such a response would be an example of geographically-induced ice sheet hysteresis (Garbe et al., 2020). Another location where such effects may have played a role is in NE Greenland, which experienced rapid and extensive deglaciation of a wide continental shelf, while NW Greenland on the opposite side of the main ice divide instead experienced slower and more limited retreat of ice margins (Fig. 21c). Consequently, during the Late-Glacial and early Holocene periods, a significant ice divide migration may have occurred in NE Greenland. The recent findings of a possible 2290 re-configuration of NEGIS during the early Holocene (Franke et al., 2022) could perhaps be linked to such regional adjustments in ice-flow.

*Influence of heterogeneous onshore bed topography*

The present-day onshore Greenland topography is spatially heterogenous, with the CE, SE, and SW regions characterised by prominent coastal mountain ranges, while other regions generally feature lower-lying beds, in some places resting below modern sea level, such as underneath NEGIS or the Jakobshavn and Humboldt ice streams, for instance (Morlighem et al., 2017). The latter can enhance margin retreat speed and magnitude, and promote outlet glacier instability (Jamieson et al., 2012). This may help explain why the CW Greenland Ice Sheet region likely experienced more retreat behind contemporary 2300 margins in response to the Holocene Thermal Maximum (Simpson et al., 2009; Larsen et al., 2015), although this remains a hypothesis.

Higher and more rugged topographies near the coast present orographic obstacles to precipitation from offshore. They promote ice accumulations sustaining coastal glaciers which can contribute ice flux to the main ice sheet, and locally 2305 decrease the impact of negative mass balances on ice-margin retreat during warming phases. Higher coastal topographies enable steeper surface slopes near ice-sheet margins, and less accumulation area loss for a given rise in Equilibrium Line Altitude during warming (Fig. 21b, scenario 2). Moreover, high coastal mountains likely promote less slippery bed conditions, with a rougher bed and more bedrock areas at higher elevations potentially enabling more cold-based conditions. These mechanisms could together explain why the SE sector of the Greenland Ice Sheet is characterised by a margin located 2310 closer to the shore than other more deglaciated regions, despite significantly warmer local atmospheric and sea surface



temperatures (Fig. 20; Osman et al., 2021). Indeed, while greater ocean and atmospheric warming likely caused this region to lose its former marine-terminating margins earlier and more rapidly than in other sectors, this high coastal topography feedback may have caused the ice sheet to stabilise when becoming predominantly land-terminating, with local margins remaining near the shore throughout the Holocene.


Generally, we observe a pattern of stabilisation of ice-sheet margins during their marine-to-terrestrial transition in several regions. Indeed, an abundance of prominent moraines and other ice-marginal landforms close to present-day coasts can often be observed where present-day topography reaches above the marine limit. This is common in the Ilulissat, Sisimiut, and Nuuk regions (e.g. the Kapisigdlit stade moraines; Young et al., 2021) (Fig. 12), for example, but also in NE Greenland,

onshore of major fjords (e.g. Danmarks Fjord). These landforms imply stillstands of the margin once floatation and calving fronts were lost, suggesting that calving and sub-shelf melt caused more ablation than subaerial processes in these regions, during the Late-Glacial to mid-Holocene retreat of the ice sheet. Here again, the role of bed topography is critical as the width, depth, and length of Fjord systems control how much margin retreat is required prior to ice becoming fully grounded. Due to the added ablation effect of calving, topographic heterogeneities between Fjords and inter-Fjord sectors can generate

different modes of deglaciation for adjacent outlet glaciers, as observed in North Greenland (Larsen et al., 2020). Therefore, the transition from marine to land-terminating margins, which likely occurred at different times in different regions and valleys, is an essential factor influencing the dynamics of former ice sheet margin retreat in Greenland. We note the spatial and temporal uncertainties of the PaleoGrIS 1.0 reconstruction are likely too large to assess such valley-scale feedback quantitatively. Moreover, current empirical reconstructions of relative sea level change in Greenland are too uncertain to

establish an accurate map of paleo sea levels at the ice sheet scale. In most Greenland regions, our ability to test the impact of this mechanism requires thorough modelling experiments (with a coupled glacial-isostatic-adjustment model) that are calibrated against observations and ran at high (<1 km) spatial resolutions.

Finally, we argue that when subglacial topography is asymmetrical on opposite sides of an ice sheet, it can cause potent ice-

divide migrations during ice-sheet build up and demise phases and lead to geographically-induced hysteresis (Larsen et al., 2016). In central Greenland, for instance, significantly higher coastal topographies (up to 2500 m a.s.l.) lie underneath and around the eastern ice-sheet margin relative to the western margin, which features a wide, low-lying subglacial drainage basin (Fig. 21). During strong cooling phases and ice build-up towards full glacial extent, the ice sheet in CE Greenland draining towards Scoresby Sund would have required thicker ice to overflow mountains and become unconstrained, than in

CW Greenland where ice feeding the Jakobshavn Ice Stream faces no such obstacle. This asymmetrical thickness build-up may have generated an ice-sheet dome increasingly skewed towards the East during advance to full glacial extent, resulting in a potent eastwards migration of the main ice divide in central Greenland. Conversely, during deglaciation, when eastern margins retreated towards or behind present-day positions and became increasingly constrained by coastal mountains impeding ice evacuation, a westward migration of the divide may have occurred in this region (as drawn in Fig. 21). In fact,



one could argue that westward migration of the main ice divide in central Greenland might continue and possibly accelerate in the future, as the ice sheet margin thins, retreats, and becomes increasingly blocked by higher topographies and reversed bed slopes to the East. We therefore speculate that ice divide migrations likely play an important role in controlling the position of ice streams and influencing ice margin retreat dynamics as noted for other paleo ice sheets (e.g. Greenwood & Clark, 2009). The influence of such mechanisms could be further tested by running high order/resolution model simulations

constrained by ice extent observations.

To summarise, we here raise the hypothesis that at the regional scale, the heterogeneous and sometimes asymmetric nature of Greenland's terrestrial and continental-shelf topographies may be responsible for:

- Skewed ice-sheet profiles in surface elevation

- Regional variability in thickness and surface slope

- Regional variability in mass-balance response to warming as a consequence of the above

- Re-arrangements of major ice streams and surface velocities during retreat

- Spatial variability in the timing of marine-to-terrestrial transitions resulting in non-synchronous margin stabilisations

- Major ice-divide migrations during both advance and retreat phases

Together, these re-arrangements may cause inter-regional variability in the speed and magnitude of Greenland Ice Sheet margin retreat during deglaciation.




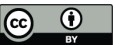



**Figure 21. Schematic model describing the hypotheses raised in section 4.2.2, which explore the potential roles of asymmetrical continental shelf- and coastal bed topographies on the observed inter-regional variability in ice-sheet margin retreat dynamics in Greenland during deglaciation. Perhaps these ideas help explain the regional variability in both rate and magnitude of retreat. The subglacial topography and bathymetry data displayed on right-hand panels is a merged product of the BedMachine v4 and GEBCO (2022 release) DEMs.**

## 5. Summary and conclusions

- We present PaleoGrIS 1.0, an ice-sheet wide reconstruction of the Late-Glacial and Holocene Greenland Ice Sheet margin evolution, documenting the deglacial back-stepping of ice margins to the present-day ice sheet edge. This new reconstruction is based on around 194,000 geomorphological and 1,450 geochronological markers of former ice extent. It yields ice marginal isochrones at a temporal resolution of 500 years and spans ~7.5 kyr from approximately 14 to 6.5 kyr BP.

- This reconstruction of the ice sheet margin evolution through time, based on our Greenland-wide mapping of ice-marginal landforms combined with a synthesis of published geochronological data, enables us to pinpoint the timing and dynamics of key glaciological events, such as the unzipping of the Innuitian and Greenland Ice Sheets and subsequent opening of Nares Strait for example, here estimated to have occurred at ~8.5-8 kyr BP.

- The PaleoGrIS 1.0 isochrones enable us to quantify the Late-Glacial and Holocene areal extent change of the entire Greenland Ice Sheet. We find that between 14-13 kyr BP until the present day, the ice sheet has lost approximately one third of its areal extent (0.89 million km$^2$). Between 14-13 and 9-8.5 kyr BP, the ice sheet experienced a near constant rate of areal extent loss of $170 \pm 27$ km$^2$ yr$^{-1}$. We also find Greenland outlet glaciers experienced faster retreat rates between ~12 and ~9 kyr BP, while both rates of areal extent loss and margin retreat decreased markedly after ~9 kyr BP. The timing of such faster retreat coincides with the most potent local climate warming to have occurred between the Younger Dryas (~13-11.7 kyr BP) and the end of the pre-industrial era (~1850 AD). In spite of local glaciological complexities, the deglacial evolution of ice margins at the ice-sheet scale is strongly correlated to, and controlled by, changes in atmospheric and oceanic temperatures. These results imply the Greenland Ice Sheet was highly sensitive to climate warming in its recent history, and will likely remain so in the future.



- We find the Greenland Ice Sheet minimum Holocene extent was reached later than ~6.5 kyr BP, which is hundreds to thousands of years after the peak warming of the Holocene Thermal Maximum. We conclude that 'inertia' in the ice-sheet system (a mix of fast and slow glaciological responses) has caused a centennial to millennial-scale time
lag in its response to the cessation of Holocene warming. Understanding and quantifying such inertia, by better calibrating numerical ice sheet models against data for instance, is of crucial importance to anticipating the magnitude of committed mass loss and the response of the ice sheet to ongoing global warming over the next few centuries.

- When quantifying retreat in different Greenland regions, we find heterogeneous responses in the magnitude and rate of the ice-sheet areal extent loss, and in outlet glacier retreat rates. While some of this heterogeneity may be related to spatial variability in climate and ocean forcing, other non-climatic factors likely play important roles in determining the regional-to-valley scale dynamics of deglacial ice margin retreat. In particular, we hypothesise that in certain Greenland regions, the asymmetrical configurations of continental shelf- and onshore bed- topographies
on opposite sides of the ice sheet may be responsible for inter-regional variability in the magnitude and timing of margin evolution, with implications for both advance and retreat phases. Ice sheet modelling experiments could usefully explore these hypothesised controls, which may continue to be relevant regarding future ice loss and sea level rise contributions.

- Despite remarkable and accelerating efforts from the empirical community to produce an extensive library of ice extent markers around the periphery of the Greenland Ice Sheet, we find vast onshore and offshore regions with sparse data, and which yield isochrones of low confidence. There is therefore much potential for improving our knowledge of deglacial Greenland Ice Sheet margin evolution by producing new robust paleo-glaciological investigations from key understudied regions. We believe the PaleoGrIS 1.0 reconstruction and database can prove
useful for identifying important study sites and motivating future onshore and offshore field activities.

- PaleoGrIS 1.0 is made available as an open-access database which may prove useful to both the empirical and ice sheet modelling communities, and for data-modelling comparison exercises. To reduce data-processing tasks and make our database usable by both communities we provide isochrone-related datasets in both GIS
(geotiff/shapefile) and NetCDF formats. The database also aims to be in a format suitable to the production of improved future versions of the reconstruction.



**Data availability.**


The PaleoGrIS 1.0 database will be made open access on the PANGAEA® Data platform. It will include the geomorphological mapping dataset (shapefile format), the geochronological compilation (2 Excel spreadsheets and numerous shapefiles for various products, including for both raw ages and summary event ages), a A0 poster of the reconstruction, and the PaleoGrIS 1.0 isochrone reconstruction in both tiff and NetCDF formats, for both the empirical and
modelling communities to use.

**Data online repository.**

The PaleoGrIS 1.0 database will be made open access on the PANGAEA® Data platform. It will be freely available online, as long as original publication is cited when used and/or referred to. The online repository is accessible using this DOI: …………

**Author contributions.**

C.D.C. conceived and guided the study. Input from J.C.E. and T.P.M.L. contributed to the design of the investigation. S.J., C.H., C.D.C., and T.P.M.L. conducted geomorphological mapping of ice marginal landforms from remote sensing (in relative order of contribution). T.P.M.L. compiled and filtered the geochronology with input from C.H.'s MSc thesis work.
T.P.M.L. created the isochrone reconstruction with feedback from C.D.C. and J.C.E. and T.P.M.L. conducted the subsequent quantitative analyses. T.P.M.L. wrote the manuscript with feedback from C.D.C. and J.C.E. primarily, and other co-authors for subsequent drafts. T.P.M.L. produced all maps, figures, tables, and prepared/formatted the PaleoGrIS 1.0 database. C.D.C, J.C.E. and C.D. helped with providing feedback and ideas on figures, along with other co-authors to a lesser extent.

**Competing interest.**

We (the authors) hereby declare that this scientific investigation presents no known competing financial interests or personal conflicts influencing research output.

**Disclaimer.**



**Acknowledgements.**

We express our gratitude towards all individuals who contributed informal ideas and feedback to this work, including Prof. Lev Tarasov, Prof. Jason Briner, Prof. David H. Roberts, and members of the PALGLAC team Remy L.J. Veness, Dr. Helen E. Dulfer, and Dr. Benjamin M. Boyes.


**Financial support.**

This study benefited from the PALGLAC team of researchers with funding from the European Research Council (ERC) under the European Union's Horizon 2020 research and innovation programme to C.D.C. (Grant Agreement No. 787263), 2525 which supported T.P.M.L., S.J., S.L.B. and C.D. J.C.E. acknowledges support from a NERC independent fellowship award (NE/R014574/1).

**Review statement.**






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
