# Peer review of "A Greenland-wide empirical reconstruction of paleo ice-sheet retreat informed by ice extent markers: PaleoGrIS version 1.0"

_Climate of the Past, 2023_

## Author Response (AR1)

Reviewer:

General comments

Overall, this a useful contribution to our overall understanding of the deglacial history of the Greenland ice Sheet (GrIS). The methods employed are tried and tested on other recent ice sheet reconstruction projects (e.g. Britice-Chrono). The mapping approach enables the broad demarcation of ice marginal positions, and in combination with pre-existing geochronological data (sourced and collated from multiple sources; e.g. Ice-D), this provides a series of isochrones with which to constrain ice retreat between 14 and 6.5 ka. This period of GrIS is already well known from multiple sectors of the GrIS, but it is useful to visual ice sheet wide retreat patterns and rates to help elucidate ice sheet response to climate forcing. The construction and provision of this database represents a serious amount of work which will make a valuable contribution to the research community.

Authors' response: We thank the reviewer for these positive comments.

1) Reviewer:

However, the latter parts of the paper that attempt to explore regional drivers of ice sheet change are weaker and require some additional critical discussion.

The lack of mention/appreciation of the sub-surface ocean temperatures in driving ice off the continental shelves undermines the discussion of ice sheet response to forcing. Atlantic Water ingress and the influence of the East and West Greenland Currents (different leads/lags pan-ice sheet) during deglaciation was pivotal in the response of the marine terminating margins to ocean heat flux. This needs acknowledging and discussing.

Authors' response: We thank the reviewer for this important point. As detailed further below in response to the relevant specific comment (number 18): we have added text in line with this comment in the "*4.2.1 Ice-sheet-wide response*" section. Regarding section 4.2.2 on regional responses, a deliberate choice was made to focus our discussion more on the ice-sheet- and sector-scale topographic heterogeneities and asymmetries, as we believe such drivers to be less mentioned by the literature than climatic and oceanic forcings, but potentially also very important. In response to the reviewer's comment, however, we have now added text to this paragraph on the importance of leads and lags in sub-surface Atlantic Water ingress across past ice-sheet margins and their possible roles in causing different ice-retreat responses:

*"Moving from ice-sheet wide to a focus on individual regions (the ice catchments of Rignot & Mouginot, 2012), we find heterogeneous retreat patterns with apparent differences in both the magnitude of regional areal extent loss and the speed of ice margin retreat (Figs. 16, 17). Spatially variable climate and ocean forcings (Fig. 20) may have caused some of these differences. For instance, the saline Irminger Current transporting warm Atlantic waters offshore South Greenland (Fig. 1), likely caused this region to experience early, more intense oceanic and atmospheric warming (Levy et al., 2020; Fig. 20c). The inflow of relatively warmer, more saline sub-surface water originating from the North Atlantic Current was likely crucial in causing sub-shelf melt near the grounding lines of marine-terminating ice-sheet margins (Knutz et al., 2011; Jennings et al., 2014). It has been argued that significant leads and lags in early deglaciation across Greenland may be partly related to the complex advection patterns of this water mass, enabled by the Irminger and West Greenland current (Fig. 1), influencing ice retreat in SE, SW and CW Greenland. Similar mechanisms are regarded to have caused the intermediate Return Atlantic current to influence ice-margin retreat across the NE and CE Greenland shelves (Hebbeln et al., 1994; Nørgaard-Pedersen et al., 2003; Hansen et al., 2022)."*

2) Reviewer:

The role of sea-level is partly mentioned but not explored in detail. The switch from a full glacial sea, up rapid uplift (and falling RSL) being particularly pivotal in grounding line instability as the ice retreated to the coast at the opening of the Holocene. The loss of ice shelves and sea-ice conditions also need a mention, though the role of both in influencing rates of deglaciation are much difficult to assess.

Authors' response: We thank the reviewer for this comment and agree that RSL change is important and likely plays a significant role in driving ice-margin response by acting on the spatio-temporal variability of ice-sheet margins transitioning from marine to fully grounded terrestrial system. Paragraph 3 of discussion section on "*Influence of heterogeneous onshore bed topography*" touches on that. In line with the reviewer's comment, we have now added text to this paragraph to make it clearer that spatially-heterogeneous RSL and GIA-induced uplift are important drivers in this case:

"*Due to the added ablation effect of calving, topographic heterogeneities between Fjords and inter-Fjord sectors can generate different modes of deglaciation for adjacent outlet glaciers, as observed in North Greenland (Larsen et al., 2020). Therefore, the transition from marine to land-terminating margins, which occurred at different times in different regions due to variabilities in uplift from glacial isostatic adjustment and thus in relative sea level change (Long et al., 2011), is an essential factor influencing the dynamics of former ice sheet margin retreat in Greenland.*"

Moreover, we have now also added a new paragraph to section 4.2.1 "ice-sheet-wide response", to mention the important role of relative sea level change on influencing the timing and nature of ice-sheet-wide retreat:

"…..*Overall, these observations suggest the Greenland Ice Sheet was highly sensitive to atmospheric and oceanic warming.*

*Relative sea level change is also likely to have influenced the timing and the linear nature of the ice-sheet-wide retreat pattern during the Late-Glacial and early-to-mid Holocene periods (Figs. 15, 16). In most coastal regions, relative sea level rose rapidly after ~16 kyr BP, reaching a spatially-variable high stand of 50-120 m at around 12-10 kyr BP (Funder & Hansen 1996; Gowan, 2023; Lecavalier et al., 2014; Simpson et al., 2009). The timing of this relative sea level high-stand therefore also coincides with the observed peak in outlet glacier retreat rates (Fig. 19). More specifically, this rapid sea level rise is thought to have promoted the early deglaciation of inter-stream sectors of the marine-terminating former ice sheet, which likely presented thinner ice (<300-400 m) grounded on shallower sections of the continental shelves, and were more vulnerable to buoyant lift-off and calving-induced retreat (Roberts et al., 2009). The importance of this mechanism has moreover been confirmed by modelling studies experimenting with different sea level forcings to nudge modelled Greenland Ice Sheet extent (e.g. Lecavalier et al., 2014; Simpson et al., 2009).*

*Based on the observation of such broadly linear response in ice extent,….*"

The next discussion section on "regional responses" is mainly focused on the heterogeneous magnitude and rate of retreat between different sectors of the ice-sheet, with a particular focus on potential topographic drivers of such heterogeneities related to asymmetries in subglacial topographies. We made the decision to focus this section on this particular point as we believe it to be more rarely mentioned than climatic and oceanic drivers, arguably more important drivers at the outlet-glacier scale, but which are also the subjects of the majority of Greenland studies. We have modified the sentence leading to the section of the discussion on regional responses, to make this point clearer:

"*However, at the regional to valley scales, we believe non-climatic and oceanic factors also help explain the variability. We discuss and hypothesise some of these mechanisms in the following sections, with a particular focus on ice-sheet- and sector-scale topographic heterogeneities and asymmetries. Whilst numerous other non-climatic factors, such as spatial heterogeneities in relative sea level and sea-ice cover change, for instance, may also play roles in controlling such regional heterogeneities, we here only focus on what we believe to be main or first-order drivers.*"

3) Reviewer:

The section on inertia requires additional thought. It touches on some relevant concepts but is too brief. This is particularly important in relation to the rate of Early Holocene ice loss and should be placed into context when considering the last 4500 yrs as Neoglacial conditions have prevailed. How do ice sheet-wide response rates to forcing differ during pronounced periods of -ve (Early Holocene) and +ve (NeoGl) mass balance? Have a look at the Young et al paper et al. (2013) paper for the Early Holocene and consider what are the implications of the Neoglacial for future GrIS mass balance change?

Authors' response: We are grateful to the reviewer for pushing us on this important point, and for helping improve our argument. We have now added more text to the paragraph in line with the reviewer's comment:

*"'Glaciological inertia' and delay in response*

*Although the Greenland Ice Sheet was highly sensitive to atmospheric and oceanic warming, specific glaciological responses to a forcing may vary over different timescales. Melting or calving can happen nearly instantaneously whereas changes to ice accumulation may take thousands of years to work through the system. Dynamic changes to flow geometry and the positions of ice divides, as well as responses to glacial isostatic adjustment may take even longer still (Rogozhina et al., 2011). Consequently, significant inertia in the system may have caused delay in the ice sheet extent response to deglacial warming. The PaleoGrIS 1.0 reconstruction suggests that the Greenland Ice Sheet margins remained more extensive than present in several regions until 7-6.5 kyr BP. The minimum Holocene extent of the entire ice sheet was thus not reached until later, most likely between 6.5 and 4 kyr BP, and as was previously suggested by numerical modelling experiments (e.g. Lecavalier et al., 2014). Over Greenland, δ18O ice core records (e.g. Alley, 2000; Buizert et al., 2018b) and temperature reconstructions from paleo-climate data assimilations (e.g. Buizert et al., 2018a; Erb et al., 2022) all suggest Holocene mean annual temperatures reached maximum values between ~9 and ~6 kyr BP, followed by gradual Neoglacial cooling until the end of the pre-industrial era (~1850 AD; Kjær et al., 2022) (Fig. 21). Therefore, our ice-sheet-wide empirical reconstruction agrees with previous observations that during the Holocene, the ice-extent response of the Greenland Ice Sheet lagged the cessation of warming, potentially by thousands of years. This delay in turn caused a lag in the response of the ice sheet to positive mass balances during mid-to-late Holocene cooling. Indeed, while both summer insolation (Berger & Loutre, 1991) and temperature proxy records (e.g. Erb et al., 2022) reveal atmospheric cooling across Greenland from 8-6 kyr BP to the pre-industrial era, significant neoglacial glacier expansion did not occur until 2.5-1.7 kyr BP (Kjær et al., 2022). Finally, the Greenland-Ice-Sheet's response to anthropogenic warming that started around 1880-1900 was also delayed by almost a century, with mass and ice-extent loss of ice-sheet-scale significance only starting in the 1980s (Kjeldsen et al., 2015; Yang et al., 2022). Interestingly, the latter decadal delay was shorter than the centennial- to millennial-scale lag in ice-sheet's response to the earlier positive mass balances associated with mid-to-late Holocene cooling, thus possibly highlighting a positive*

*correlation between the duration/magnitude of the climate forcing and the length of the delay in subsequent ice-extent response. However, such a lagged response observed at the ice sheet scale was not ubiquitous across the entire ice-sheet margin, as certain more dynamic outlet glaciers appear to be more well-coupled with short-lived early Holocene temperature changes (e.g. Jakobshavn Isbræ; Young et al., 2013b). Centennial to millennial-scale inertia of the Greenland Ice Sheet following warming has been previously suggested (e.g. Yang et al., 2022), and has important implications for ongoing climate warming because it results in committed future mass losses, and thus sea level rise contributions, that may last for centuries to millennia (e.g. Greve & Chambers, 2022)."*

4) Reviewer:

Finally, the discussion Greenland ice streams with respect to their dynamic behaviour though time. The designation of an 'isbrae' type ice stream originated in Greenland. Selective linear erosion and topographic over-deepening have controlled valley/fjord evolution of Quaternary timescales. Topographic control of ice stream position and behaviour through time is particularly critical as the ice steps back from mid/inner shelf positions into the coastal fjords. So, topography becomes pivotal in controlling ice sheet/ice stream dynamics in the window 12-8ka. Caution is required when linking ice stream behaviour to climate forcing. On the mid/outer continental shelves, 'soft-beds' and low gradient seafloors may have provided better conditions for ice stream switching/re-arrangement earlier in the glacial cycle but we know little of such history as yet.

Authors' response: We thank the reviewer for this interesting point. The hypothesis of potential ice stream reconfigurations during deglaciation is only briefly mentioned in our discussion: which in the relevant paragraph focuses more on the idea of ice-divide migrations as a result of possibly different ice sheet geometries during full glacial extent. Such migrations may in turn have modified the configuration of ice streams when comparing ice sheet velocity fields at full glacial extent (e.g. 17 kyr BP) with their configurations towards the early Holocene, when indeed coastal topographies likely became more important in controlling ice stream position by constraining ice flow. Whilst individual ice streams may respond at different times and appear highly influential as individuals, in actual fact they might be stealing or giving ice flux away to adjacent catchments (whose boundaries migrate), such that it all gets smoothed out at a regional or ice-sheet scale. This was a key finding for example for ice streams of the Laurentide Ice Sheet (Stokes et al. 2016) where ice stream activity was found to scale to ice sheet volume.

In line with the reviewer's comment: we have added text to the third paragraph in the discussion section on "*Influence of heterogeneous continental-shelf topographies*":

*"Consequently, during the Late-Glacial and early Holocene periods, a significant ice divide migration may have occurred in NE Greenland. The recent findings of a possible re-configuration of NEGIS during the early Holocene (Franke et al., 2022) could perhaps be linked to such regional adjustments in ice-flow. Around Greenland more generally, the ~12 to ~8 kyr BP window may have been critical for ice-stream re-adjustments, as the ice sheet margin stepped back and thinned from mid/inner shelf positions into more topographically confined coastal fjords. Such confinement increased the fixing in position and channelising of ice flow, and pre-existing overdeepenings may have induced some stepped behaviour in retreat. The early Holocene interval was thus likely associated with a stronger topographic control on ice stream position and behaviour throughout the Greenland Ice Sheet (Roberts et al., 2009)."*

Regarding this comment: *"Caution is required when linking ice stream behaviour to climate forcing."*: we fully agree with the reviewer. There is however no linking of ice stream behaviour to climate forcing in our discussion, since we decided to explicitly focus this section of the discussion on non-climatic drivers of ice-sheet extent change.

**Specific comments**

1) **Reviewer:**

Abstract:

This is a reconstruction based on pre-existing geochronological and geomorphic data with new ice margin positions mapped.

Authors' response: We assume this comment refers to the statement in the abstract about PaleoGrIS being: *"the first Greenland-wide isochrone reconstruction of ice-sheet extent evolution through the Late-Glacial and early-to-mid Holocene informed by both geomorphological and geochronological markers"*.

We thank the reviewer for pointing out this sentence can be confusing to the reader, this early in the manuscript. We have therefore modified the sentence, so it reads:

*"we here present PaleoGrIS 1.0, a new Greenland-wide isochrone reconstruction of ice-sheet extent evolution through the Late-Glacial and early-to-mid Holocene informed by both geomorphological and geochronological markers."*

2) **Reviewer:**

Li 78: *starting from the wrong state* – reword

Authors' response: The sentence was reworded to: *"As few models include the committed response to late-Quaternary environmental changes, current projection simulations might effectively be starting from a possibly unrealistic state, whose influence is thus far unquantified."*

3) Reviewer:

Li 85: *and pattern of ice margin retreat during the Late-Glacial and early-to-mid Holocene periods (between ~14 and ~6 kyr BP) remains poorly constrained*. This is a misleading statement. The central west coast (Disko and Uummannaq regions) has a well constrained deglacial history. As do many areas in the S, SE, SW, E, NO and NE.

Authors' response: Apologies, the sentence is indeed misleading in its current form. The take home message being empirical constraints are by nature highly heterogeneous: we have now modified the sentence to: "… *remains heterogeneously constrained in numerous regions*."

4) Reviewer:

Li 88: Was the GriS influenced significantly Meltwater Pulse 1A? Where is the evidence? Elaborate.

Authors' response: We thank the reviewer for pointing this out. The example of Meltwater Pulse 1A was originally cited to mention one episode of rapid Relative Sea Level change within the time frame studied here. However, we agree this can confuse the reader as one might expect a causation between this specific event and GrIS margin evolution. This would not suit an introduction and is too speculative. We have now removed this cited example from the sentence.

5) Reviewer:

Li 92: *A key question concerns how far the ice sheet retreated behind its contemporary margin in response to the warming of the Holocene Thermal Maximum (10-5 kyr BP; Cartapanis et al., 2022)*, You need to add more references here. This is a question that has been asked way before 2022, by many, many authors.

Authors' response: We thank the reviewer for this comment and have added Weidick *et al*. (2004) to this bracket.

6) Reviewer:

Li 97: Again, a lot of references missing a lack of appreciation of many studies from the last 50yrs that have built our understanding of GrIS retreat over the last 15,000yrs. Acknowledge some of the pioneers, not just recent review papers!

Authors' response: We have added (Ten Brink & Weidick, 1974; Hjort, 1979; England, 1985) to the bracket.

7) Reviewer:

Li 103/4: *facilitate targeting of understudied regions for future fieldwork*.  Why do you think there are understudied parts of Greenland? I understand the statement, but the inference is 'the community' have left holes, when of course this is an issue related to access and logistics. This does the Greenland research community a bit of dis-service.

Authors' response: We fully agree with the reviewer that the main reason for specific regions being less well studied is naturally access and logistics. The sentence "*this work will facilitate targeting of understudied regions for future fieldwork*" was not originally meant to infer that the community has deliberately left holes, but rather that there remains areas to be studied, and we are sorry that it might have been interpreted that way. We have now rephrased to:

"*It would enable ice-sheet scale analyses, help keep a standardised record of dating investigations, facilitate targeting new study regions for future fieldwork, and provide a reconstruction for calibrating ice sheet model simulations.*"

Moreover, we have now removed the term "understudied" from the full manuscript and rephrased relevant sentences accordingly.

8) Reviewer:

Li 175: Contradicts your earlier statement about a lack of work.

Authors' response: we have now modified the prior statement which we agree was misleading. See response to comment above.

9) Reviewer:

Li 225: *and unlikely to be preserved, we consider our mapped ice-marginal landforms to have been deposited during the last deglaciation*…. Why make this assumption? This is a total unknown in Greenland. If you ran a similar exercise in Iceland, you would find that overridden moraines are extremely common in the geological record. In Greenland, there may not be as many examples of overridden moraines, but there are places where moraine spacing gets condensed due to offshore/onshore transition, topographic  and climate control. So, the rates of ice margin change are extremely complex. Have a look at the extensive literature on the fjord stade moraines in Disko Bugt (See Young et al. 2013). This is a clear example of both ice

stream and interstream areas feeling a rapid climate downturn (9.3. and 8.2 events) but also being sensitive to ice flux and topography.

Authors' response: We thank the reviewer for this point. This assumption, based only on the low quantity of observations and dating of pre-LGM ice-marginal deposits in Greenland, is indeed hypothetical and was likely too confidently asserted.

To stress that this assumption in imperfect but our only option for this ice-sheet-wide exercise: we have now modified this part of the paragraph to:

"*Due to lack of pre-LGM ice margin chronologies across Greenland, and following the assumption that glaciogenic deposits relating to previous glaciations were overridden during the last glaciation (Funder et al., 2011) and thus less likely to be preserved (with exceptions: i.e. Mejdahl & Funder, 1994; Kelly et al., 1999), we adopt the assumption that our mapped ice-marginal landforms were deposited during the last deglaciation, between ~17 kyr BP and present.*"

10) Reviewer:

Fig 2: the Maps of ice marginal landforms look great.

Li 337: *which given the young nature of the ages (<14 kyr BP) and the relatively low magnitude of surface elevation change related to glacial isostatic adjustment during the Holocene*, This is misleading. In many parts of the ice sheet there has been 50-120m of uplift postglacially. The authors need to engage with the sea-level literature.

Authors' response: We fully agree concerning the reported magnitude of GIA-related uplift post-glaciation in Greenland. However, this magnitude of uplift (50-120 m) is indeed relatively low when compared to many other deglaciated regions of the world (e.g. Canada, West Antarctica, Norway, Sweden, Finland...: where uplift has been > 400 m in numerous places), and is relatively low specifically when investigating the sensitivity of TCN exposure ages to surface elevation change through time. We refer the reviewer to the paper of Jones et al. (2019) on this matter and we here copy the response to reviewer 1 on this same point:

"We agree that this [the spatially-variable impact of GIA uplift on TCN age correction] will be important to re-assess once a publicly-available sea level indicator and proxy database exists. However, we note that recent assessments of the impact of GIA-related uplift on cosmogenic exposure ages in Greenland (i.e. Jones et al., 2019) show that ages would have to be corrected at maximum (moreover only in SW Greenland) by ~7%, which, for Holocene ages (vast majority of deglacial ages in Greenland), would not represent a correction in the order of thousands of years, but rather in the order of a few hundred years. The more important factor to consider for TCN ages is: when did the uplift happen, and how quickly? This can be

harder to determine. If most of the rebound had already occurred prior to, or towards the onset of the exposure period (say early Holocene), then the impact of GIA correction on the resulting exposure age would be minimal and well below analytical uncertainties."

In order to make our point clearer here, we have added the range of magnitude of uplift in Greenland to the sentence to better indicate that indeed, such magnitude of surface elevation change is relatively low when specifically assessing its impact on TCN ages:

*", which given the young nature of the ages (<14 kyr BP) and the relatively low magnitude of surface elevation change (50-120 m) related to glacial isostatic adjustment during the Holocene, is not thought to cause age offsets greater than analytical uncertainties in most Greenland regions (Jones et al., 2019)."*

11) Reviewer:

Li 884: *drop significantly between 6.5 kyr BP and 2-1 kyr BP, when the ice sheet was responding to the Holocene*. This needs rewording. Post the HTM the ice sheet reached its minima sometime between 7 and 5 ka probably. By 5 – 4.5 ka we then we see the onset of the Neoglacial and ice sheet re-expansion culminating in the LIA.

Authors' response: We thank the reviewer and have modified this part of the paragraph to reword it like so:

*"Following the Holocene Thermal Maximum, between 6.5 kyr BP and 2 kyr BP, the number of available dates drop significantly (Fig. 7). This coincides with the ice sheet reaching its minimum Holocene extent (between ~7 and ~4.5 kyr BP), which in most Greenland regions was either as extensive, or more retreated than today's margin position (Larsen et al., 2015; Briner et al., 2016). After ~5-4 kyr BP, the onset of Neoglacial cooling caused ice-sheet re-advances culminating in the Little Ice Age, which explains the small relative increase in the number of dates in our compilation after 2 kyr BP (Fig. 7)."*.

Please note we use slightly more conservative time ranges as suggested as we are here referring to the entire ice sheet margin and that the precise timing of minimum ice-sheet extent during the Holocene is both mostly unknown, and likely spatially heterogeneous.

12) Reviewer:

*or more retreated around most of its perimeter* – reword

Authors' response: We have done so accordingly, see response directly above.

13) Reviewer:

Li 780: *For TCN exposure dating, misleading ages can result from nuclide inheritance causing too-old apparent ages, postdepositional disturbance causing too-young apparent ages, or laboratory contamination/errors potentially causing both (Dunai, 2010).* But more often than not these TCN ages have already been discussed as part of the original research publication.

Authors' response: We thank the reviewer for this comment which we fully agree with, hence why our identification of outliers in the database only occurs when described as such by original authors, without further interpretation, as is mentioned in the methods section (see line 361).

In line with this comment, we have added this sentence here:

*"For TCN exposure dating, misleading ages can result from nuclide inheritance causing too-old apparent ages, post-depositional disturbance causing too-young apparent ages, or laboratory contamination/errors potentially causing both (Dunai, 2010). Such issues are in most cases reported in original publications enabling us to easily identify outliers."*

14) Reviewer:

Li 800: Limitations – these are covered in an open and clear manner.

Li 1640. There is no mapped reconstructions for the Uummannaq region – why - too complicated ?

Authors' response: Following this comment, we have now created a new 22nd figure (Fig. 14) in the same style as the figures describing the reconstruction in other regions, and appended the text accordingly. Please see section 3.2.7. for a description of our reconstruction and the empirical data it is based on in this specific region.

15) Reviewer:

Fig 14 is a useful summary to visualise this work.

Li 1904: Regional intercomparisons. This is useful. Latitudinal influence is not surprising.

Retreat rate calculation and comparisons are useful. Comparison with air and ocean temp records are useful.

Li 2100: I think the community already know where the holes in the data are!

Authors' response: We agree this must certainly be the case for researchers and teams with ample experience in investigating Greenland's past evolution. Nonetheless, we believe it remains useful to clearly present and highlight areas with knowledge gaps, in particular to researchers new to this field, or with little experience in Greenland. Figure 20 (with data gaps highlighted) was presented at two distinct conferences in 2023. Both times, experienced Greenland researchers engaged specifically with that figure, and wanted to have a closer look at it after the presentations. It thus seems to be of interest to members of the empirical community.

16) Reviewer:

Li 2115: *It must be noted that such data gaps are most likely related to the increased logistical, safety, access, and financial constraints associated with field investigations in more remote and problematic regions. Furthermore, a great source of uncertainty impeding a better understanding of the deglacial dynamics originates from the ice sheet being generally more extensive today than between ~6 and ~2 kyr BP, when responding to the Holocene Thermal Maximum*. This is just stating the obvious.

Authors' response: The sentence: *"it must be noted that …."* Has now been removed from this paragraph.

17) Reviewer:

Li 2161: *of a more-retreated-than present* – reword

Authors' response: The sentence was reworded to: "The findings of former ice-sheet margins likely located behind present-day ones during and following the Holocene Thermal Maximum (Briner et al., 2014),"

18) Reviewer:

2145 *4.2.1 Ice-sheet-wide response* -  This section is dealt with very briefly and lacks a critical appraisal of ice/atmos/ocean interactions and feedbacks. The linear relationships between air and sea surface temp need exploring in more detail, but the complete lack of recognition of the role of sub-surface ocean temps in driving ice off the continental shelves really undermines this discussion. Atlantic Water ingress

and the behaviour of the East and West Greenland Currents during deglaciation and the onset of Holocene was pivotal in the response of the marine terminating margins to ocean heat flux. This point needs to be addressed You touch on this lines 2245 – 2255 but it's very brief.

Authors' response: We thank the reviewer for this pointing this out. To add more details concerning the essential role of ocean warming and subsurface temperature increases in driving mass loss and ice retreat during deglaciation and the early Holocene, we have now added new text to the paragraph:

"....*These correlations suggest climate and ocean forcing were the dominating agents of former ice-sheet extent change. For instance, the overall acceleration of outlet-glacier retreat that we observe between ~11.5 and ~9.5 kyr BP is coeval with the potent atmospheric warming that characterised the Younger-Dryas-to-early-Holocene transition in the North Atlantic and Greenland regions (Grootes et al., 1993; Fig. 20). This faster retreat also coincides with significant ocean warming, with marine records indicating increasing sea surface temperatures offshore most Greenland regions from ~16 kyr BP, and reaching maximum warmth at ~10-8 kyr BP (e.g. Williams, 1993; Jennings et al., 2006; 2017; Osman et al., 2021: Fig. 20). More particularly, offshore SE, SW, and CW Greenland regions, warmer and saline sub-surface Atlantic water ingress from the Irminger current (Fig. 1) is thought to have caused rapid mass loss and initial retreat of grounded ice margins from the outer continental shelves (Knutz et al., 2011; Ó Cofaigh et al., 2013; Jennings et al., 2017). Similar forcing is thought to have occurred in NE and CE Greenland, with warmer saline Atlantic water advected southward from Fram Strait, along the Greenland coast, and across continental shelves to marine terminating margins, with the establishment of the West Spitsbergen, Return Atlantic and East Greenland currents during deglaciation (Hopkins, 1991; Hebbeln et al., 1994). Furthermore, ...*"

19) Reviewer:

Li 2172: The section on inertia doesn't really say or prove anything. On a Holocene timescale how does the mass balance regime of the GrIS change with respect to long term inertia and ice sheet response times?

Authors' response: We are grateful to the reviewer for pushing us on this point, and helping improve our argument. We have added text to the paragraph in line with the reviewer's comment.

*"'Glaciological inertia' and delay in response*

*Although the Greenland Ice Sheet was highly sensitive to atmospheric and oceanic warming, specific glaciological responses to a forcing may vary over different timescales.*

*Melting or calving can happen nearly instantaneously whereas changes to ice accumulation may take thousands of years to work through the system. Dynamic changes to flow geometry and the positions of ice divides, as well as responses to glacial isostatic adjustment may take even longer still (Rogozhina et al., 2011). Consequently, significant inertia in the system may have caused delay in the ice sheet extent response to deglacial warming. The PaleoGrIS 1.0 reconstruction suggests that the Greenland Ice Sheet margins remained more extensive than present in several regions until 7-6.5 kyr BP. The minimum Holocene extent of the entire ice sheet was thus not reached until later, most likely between 6.5 and 4 kyr BP, as was previously suggested by numerical modelling experiments (e.g. Lecavalier et al., 2014). Over Greenland, δ18O ice core records (e.g. Alley, 2000; Buizert et al., 2018b) and temperature reconstructions from paleo-climate data assimilations (e.g. Buizert et al., 2018a; Erb et al., 2022) all suggest Holocene mean annual temperatures reached maximum values between ~9 and ~6 kyr BP, followed by gradual Neoglacial cooling until the end of the pre-industrial era (~1850 AD; Kjær et al., 2022) (Fig. 21). Therefore, our ice-sheet-wide empirical reconstruction agrees with previous observations that during the Holocene, the ice-extent response of the Greenland Ice Sheet lagged the cessation of warming, potentially by thousands of years. This delay in turn caused a lag in the response of the ice sheet to positive mass balances during mid-to-late Holocene cooling. Indeed, while both summer insolation (Berger & Loutre, 1991) and temperature proxy records (e.g. Erb et al., 2022) reveal atmospheric cooling across Greenland from 8-6 kyr BP to the pre-industrial era, significant neoglacial glacier expansion did not occur until 2.5-1.7 kyr BP (Kjær et al., 2022). Finally, the Greenland-Ice-Sheet's response to anthropogenic warming that started in the early 1880-1900 was also delayed by almost a century, with significant mass and ice-extent loss only starting in the 1980s (Kjeldsen et al., 2015; Yang et al., 2022). Interestingly, the latter decadal delay was shorter than the rather centennial- to millennial-scale lag in ice-sheet's response to positive mass balances associated with mid-to-late Holocene cooling, thus possibly highlighting a correlation between the duration/magnitude of the climate forcing and the length of the delay in subsequent ice-extent response. However, such lagged response observed at the ice sheet scale was not ubiquitous across the entire ice-sheet margin, as certain more dynamic outlet glaciers were well coupled with short-lived early Holocene temperature changes (e.g. Jakobshavn Isbræ; Young et al., 2013b). Centennial to millennial-scale inertia of the Greenland Ice Sheet following warming has been previously suggested (e.g. Yang et al., 2022), and has implications for ongoing climate warming generating committed future mass losses, and thus sea level rise contributions, that may last for centuries to millennia (e.g. Greve & Chambers, 2022)."*

20) Reviewer:

Li 2182: You need to mention Kjaer et al. (2022) in relation to the Neoglacial. The Neoglacial was an ice sheet wide response to N. hemisphere cooling.

Authors' response: We thank the reviewer for this comment and have now added this reference to two locations of this paragraph.

21) Reviewer:

Li 2258 – 2267: *Ice advance across the continental shelves 18 – 15ka due to H1?* This completely unsubstantiated and needs to be amended. Most regions were retreating from the shelf edge in this window. By all means doube-check the literature (e.g. Ó Cofaigh et al., (2013), but is definitely not true for the NE. The GriS may have experienced marginal oscillation during overall retreat (e.g. GZW formation) but not climate driven advances.

Authors' response: We thank the reviewer for pointing this out, this sentence is indeed misleading and needs changing, as the timing of advance and retreat was in fact very different in different regions. We have thus changed the sentence to:

"*When the Greenland Ice Sheet reached its maximum extent (between 24 and 16 kyr BP depending on regions; Funder et al., 2011), and when local atmospheric- and more importantly sea surface- temperatures were lowest (Grootes et al., 1993; Osman et al., 2021), the ice sheet's margins had advanced across continental shelves (e.g. CW Greenland: Ó Cofaigh et al., 2013).*"

22) Reviewer:

Li 2280: *Warmings* – reword to warm periods.

Authors' response: Thank you, the sentence was reworded accordingly.

23) Reviewer:

Li 2285: *Such a response would be an example of geographically-induced ice sheet hysteresis?* This needs a fuller explanation if you want to push this idea. Topographic and geological influences come into this equation, plus ice stream v interstream areas. You mention ice divides and ice stream switch on and off. All relevant, but all fairly conjectural. The Uummannaq Ice stream 'system' is a particularly good example of the complexity of a hysteresis across one catchment.

Authors' response: We agree with the reviewer that the mention of the term hysteresis, which arguably requires more justification here, is not necessary. We have removed this sentence from both this paragraph and paragraph 4 of section "*Influence of heterogeneous onshore bed topography*".

24) Reviewer:

Li 2297: The Holocene minima in W Greenland was largely a response to surface mass balance change and a low gradient ice surface. Perceived wisdom has not related this to the ice sheet being below sea-level. Particularly the interstream regions.

Authors' response: We thank the reviewer for this insight and have now removed this sentence from this paragraph:

*"This may have contributed to the CW Greenland Ice Sheet region experiencing more retreat behind…."*

25) Reviewer:

Li 2330: *Moreover, current empirical reconstructions of relative sea level change in Greenland are too uncertain to establish an accurate map of paleo sea levels at the ice sheet scale.* You need to rethink this. There are several areas of the GriS that have superb sea-level records. The W and SW for example. There is an overview paper by Long et al. (2011) that can shed some light.

Authors' response: We thank the reviewer for this comment, this sentence was indeed misleading as such. To better articulate what was meant, i.e. that knowing where the past sea level was through time, at the valley scale, to assess the precise timing of transition from marine to terrestrial ice-sheet margins throughout Greenland, remains a challenge, we have reworded to:

*"Moreover, whilst numerous regions feature robust paleo sea-level records (e.g. the Disko Bugt area; Long et al., 2011), it remains a challenge to establish an accurate, time-dependent, and Greenland-wide map of paleo sea levels at the valley scale (Gowan, 2023)."*

26) Reviewer:

Li 2355: Ice stream switching/re-arrangements. The new work by Frenke et al. in NE Greenland has raised the possibility of recent ice stream migration/shutdown, but I wonder how important that is during deglaciation when ice thins and becomes trapped within over-deepened topography. West Greenland in particular is underlain by a 'hard' bed with valley networks established and over-deepened on million yr timescales. Ice flow corridors are, hence, largely controlled by topography – 'isbrae' type is ice streams being typical of Greenland. During deglaciation topographic control of ice dynamics is very important (e.g. Lane et al, 2014).  That's not to say ice

stream switching/hosing never happens. There is evidence for this on the continental shelves (particularly during the Miocene)

Authors' response: We thank the reviewer for this comment and interesting thoughts, and agree that near the contemporary ice-sheet margins, ice streaming is likely to be more controlled by subglacial topography given the high magnitude of overdeepening in most Greenland Fjords and coastal topographies. However, as we move upflow and away from the ice margin, ice thicknesses increase rather rapidly, and ice flow can become unconstrained by topographies. Towards the onset of deglaciation, when the ice sheet was still predominantly marine-terminating with margins towards the outer continental shelves, present-day coastal topographies would have been covered by thick ice in numerous regions. It has been hypothesized that during the full glacial extent, many contemporary ice streams and outlet glaciers currently nested within the fjord heads of the present coast may have converged and flowed on to the continental shelf as larger, composite ice streams (Roberts and Long, 2005; Weidick and Bennike, 2007). We believe it is possible to have had different ice stream configurations than present in several cases then, and subsequent reconfigurations when thinning near the margins occurred during deglaciation, as subglacial topographies increasingly constrained ice flow. Please see reply to main reviewer comment on this same point above, and see the new sentences added to discussion paragraph on the increasingly important role of topography for ice-stream re-adjustments during the 12-8 kyr BP window.

*"Around Greenland more generally, the ~12 to ~8 kyr BP window may have been critical for ice-stream re-adjustments, as the ice sheet stepped back from mid/inner shelf positions into coastal fjords and thinned towards its margins, and as pre-existing overdeepenings became increasingly important in channelising ice flow. The early Holocene interval was thus likely associated with a stronger topographic control on ice stream position and behaviour throughout the Greenland Ice Sheet (Roberts et al., 2009)."*

To introduce more nuance, we have also modified the bullet point discussion phrase slightly and rephrased it to:

*"...may be responsible for:*

*-Re-arrangements of certain ice streams and of surface velocity fields during retreat"*